# The "SafeSpace" Database of ULF Power Spectral Density and Radial Diffusion Coefficients : Dependencies and application to simulations

Christos Katsavrias[1], Afroditi Nasi[1], Ioannis A. Daglis[1,2], Sigiava Aminalragia-Giamini[1,3], Nourallah Dahmen[4], Constantinos Papadimitriou[1,3], Marina Georgiou[1], Antoine Brunet[4], and Sebastien Bourdarie[4]

[1]Department of Physics, National and Kapodistrian University of Athens, Greece
[2]Hellenic Space Center, Athens, Greece
[3]Space Applications and Research Consultancy (SPARC), Athens, Greece
[4]ONERA/Department of Space Environment, Toulouse, France

**Correspondence:** Christos Katsavrias (ckatsavrias@phys.uoa.gr)

**Abstract.** Radial diffusion has been established as one of the most important mechanisms contributing to both the acceleration and loss of relativistic electrons in the outer radiation belt, as well as to the supply of particles to the inner radiation belt. In the framework of the SafeSpace project we have used 9 years (2011–2019) of multi-point magnetic and electric field measurements from THEMIS A, D and E satellites to create a database of radial diffusion coefficients ($D_{LL}$) and Ultra Low Frequency (ULF) wave Power Spectral Densities (PSD) spanning an L* range from 3 to 8. In this work we investigate the dependence of the $D_{LL}$ on the various solar wind parameters, geomagnetic indices and coupling functions, as well as the L-shell, during the solar cycle 24. Moreover, we discuss the uncertainties introduced on the estimation of $D_{LL}$ time-series by the partial azimuthal coverage provided by in-situ measurements. Furthermore, we investigate via a superposed analysis, the dependence of the $D_{LL}$ on solar wind drivers. We show, for the first time to our knowledge, that the Interplanetary Coronal Mass Ejections (ICME) driven disturbances accompanied by high solar wind pressure values combined with intense magnetospheric compression can produce $D_{LL}^B$ values comparable or even greater than the ones of $D_{LL}^E$. This feature cannot be captured by semi-empirical models and introduces a significant energy dependence on the $D_{LL}$. Finally, we show the advantages of using $D_{LL}$ time-series by means of numerical simulations of relativistic electron fluxes performed with the Salammbô code and significant deviations in the predictions of several semi-empirical models depending on the level of geomagnetic activity and L-shell.

## 1 Introduction

The outer radiation belt consists of electrons at energies from a few hundred keV to several MeV (Daglis et al., 2019). Radial diffusion has been established as one of the most important mechanisms that contributes to this broad energy range of electrons since it can lead to both energization (Jaynes et al., 2015; Li et al., 2016; Katsavrias et al., 2019a; Nasi et al., 2020) and loss of relativistic electrons (Morley et al., 2010; Turner et al., 2012a; Katsavrias et al., 2015, 2019b).

Ultra-Low Frequency (ULF) waves in the Pc4-5 band (1–25 mHz) can violate the third adiabatic invariant L* of the energetic electrons. This drives radial diffusion by conserving the first two adiabatic invariants under the drift resonance condition $\omega = m\omega_d$, where $\omega$ is the wave frequency, $m$ is the azimuthal wave mode number and $\omega_d$ is the electron drift frequency (Elkington et al., 2003). Most often radial transport is described as a stochastic process; the result of incoherent transport of particles by electromagnetic fields that vary irregularly on time scales of the drift period of radiation belt electrons (of the order

of minutes). The radial diffusion coefficient, $D_{LL}$, has been defined to represent the mean square change of L* for a large number of particles over time.

Currently there are two widely used formalisms in order to derive radial diffusion coefficients. Falthammar (1965) distinguished the contribution of single-mode fluctuations in Earth's magnetic field and induced electric fields ($D_{LL}^M$) and perturbations in convection electric fields ($D_{LL}^E$) to derive a mathematical formulation for $D_{LL}$. On the other hand, Fei et al. (2006)

included the contributions from all azimuthal wave modes. Nevertheless, the latter authors, made the additional assumption that the magnetic field perturbations and the inductive electric field perturbations are independent, something that assumes that the two perturbations are uncorrelated. As discussed by Lejosne (2019), such an assumption is inconsistent with Faraday's law ($\nabla x \vec{E} = -\frac{\partial \vec{B}}{\partial t}$).

Specifically, Fei et al. (2006) assumed radial diffusion coefficients as the sum of the effects of perturbations in the azimuthal

electric field and the parallel magnetic field:

$$D_{LL} = D_{LL}^B + D_{LL}^E \tag{1}$$

These two components of the radial diffusion coefficients are given by:

$$D_{LL}^B = \frac{\mu^2 L^4}{8q^2\gamma^2 B_E^2 R_E^4} \cdot \sum_m m^2 P_m^B (m\omega_d) \tag{2}$$

$$D_{LL}^E = \frac{L^6}{8B_E^2 R_E^2} \cdot \sum_m P_m^E (m\omega_d) \tag{3}$$

where $\mu$ is the first adiabatic invariant, L is the Roederer's L*, q is the charge of the diffused electrons, $\gamma$ is the Lorentz factor, $R_E$ is Earth's radius and $B_E$ is the strength of the equatorial geomagnetic field on the Earth's surface. Moreover, P corresponds to the wave power at a specific drift frequency ($\omega_d$) for all the azimuthal mode numbers (m). Note that $D_{LL}^B$ includes contributions only from the magnetic field oscillations, while $D_{LL}^E$ contains contributions from the total (inductive and convective) electric field.

It is clear, from the aforementioned formulation, that in order to have accurate calculations of the radial diffusion coefficients we need accurate magnetic and electric field measurements, which of course, are not always available. To that end, efforts have been devoted to provide empirical relationships of $D_{LL}$ for radiation belt simulations, parameterizing the diffusion coefficients by the Kp index and L* parameter. These empirical models have the advantage of providing estimations/predictions of the

| Model | $D_{LL}$ Formulation [1/days] | Limitations |
|---|---|---|
| Brautigam and Albert (2000) | $D_{LL}^{EM}[BA] = 10^{(0.506 \cdot Kp - 9.325)} \cdot L^{10}$ | 0<Kp<6 |
| | | 3<L*<6.6 |
| Boscher et al. (2018) | $D_{LL}^{EM}[BOS] = 10^{(0.45 \cdot Kp - 8.985)} \cdot L^{10.2}$ | 0<Kp<6 |
| | | 3<L*<6.6 |
| Liu et al. (2016) | $D_{LL}^{E}[LIU] = 1.115 \cdot 10^{-6} \cdot 10^{(0.281 \cdot Kp)} \cdot L^{8.184} \cdot \mu^{-0.608}$ | 0<Kp<5 |
| | | 4.5<L*<7 |
| Ozeke et al. (2014) | $D_{LL}^{B}[OZ] = 6.62 \cdot 10^{-13} \cdot 10^{(-0.0327 \cdot L^2 + 0.625 \cdot L - 0.0108 \cdot Kp^2 + 0.499 \cdot Kp)} \cdot L^8$ | 0<Kp<6 |
| | $D_{LL}^{E}[OZ] = 2.16 \cdot 10^{-8} \cdot 10^{(0.217 \cdot L + 0.461 \cdot Kp)} \cdot L^6$ | 1<L*<7 |
| Ali et al. (2016) | $D_{LL}^{B}[ALI] = exp^{(-16.253 + 0.224 \cdot Kp \cdot L + L)}$ | 0<Kp<5 |
| | $D_{LL}^{E}[ALI] = exp^{(-16.951 + 0.181 \cdot Kp \cdot L + 1.982 \cdot L)}$ | 3<L*<5.5 |

**Table 1.** Widely used semi-empirical models for the estimation/prediction of the radial diffusion coefficients, their mathematical formulation, trained datasets and limitations.

$D_{LL}$ without the dependence on the in-situ measurements. Nevertheless, it is also obvious (see also table 1) that the use of a single input parameter (Kp index) is an over-simplification for a complex process such as the radial diffusion of electrons. Moreover, Kp is a global geomagnetic index, which is a proxy for the global changes in the geomagnetic field (Mayaud, 1980). On the other hand, two of the most important (external) sources for ULF waves are a) solar wind pressure pulses and b) Kelvin-Helmholtz instabilities powered by the increased solar wind speed (Claudepierre et al., 2008). Since the Kp index does not present significant correlation with either of these two solar wind parameters, it cannot account well for the mechanism of radial diffusion that enhances or depletes the electron population in the outer radiation belt.

In addition, the observed $D_{LL}$ have been shown to be highly event-specific (Jaynes et al., 2018) and physics-based models, such as the Versatile Electron Radiation Belt, cannot simulate the dynamics of the outer radiation belt observed during every storm using these empirically estimated coefficients (Drozdov et al., 2021). Several case studies have demonstrated deviations of the event-specific diffusion coefficients from the Kp-parameterized models. The recent study of Liu et al. (2018) suggests that the difference between the various models is negligible for low levels of geomagnetic activity at an equatorial distance of L-shell = 7.5 $R_E$ but can be orders of magnitude different at high levels of geomagnetic activity. At the same extent, Olifer et al. (2019) observed that, during the March 2015 geomagnetic storm, $D_{LL}^{B}$ was consistently underestimated and $D_{LL}^{E}$ was consistently overestimated by the empirical model of Ozeke et al. (2014). Furthermore, the magnitude of mis-estimation varied throughout the event and, at times, the difference between empirically modelled values and time-series of diffusion coefficients was multiple orders of magnitude.

In this work we present a new database of ULF power spectral density (PSD) and the derived radial diffusion coefficients, which has been developed in the framework of SafeSpace project funded by Horizon 2020. The SafeSpace project aims at advancing space weather nowcasting and forecasting capabilities and, consequently, at contributing to the safety of space assets through the transition of powerful tools from research to operations. To that end, a database of radial diffusion coefficients

derived from in-situ magnetic and electric field measurements, coupled with solar wind and geomagnetic parameters, as well as the accompanied analysis, is of outmost importance, not only for statistical purposes but also, for any future efforts to develop accurate models for nowcasting/forecasting the $D_{LL}$. The rest of this paper is organized as follows: section 2 describes the datasets used as input in the $D_{LL}$ database as well as the roadmap towards its creation, section 3 reports statistics which are important for future modelling efforts and section 4 presents examples of the importance of the use of $D_{LL}$ time-series in radiation belt simulations.

## 2   ULF PSD and $D_{LL}$ database

### 2.1   Data and methods

The radial diffusion coefficients were calculated directly from in-situ measurements using the approach based on the Fei et al. (2006) formulation. As mentioned before, this approach considers the compressional component of the magnetic field and the toroidal component of the electric field. To that end, we used 4-sec resolution measurements of the magnetic field vector from the THEMIS A, D and E fluxgate magnetometers (Auster et al., 2008) as well as electric field measurements from the EFI instrument (Bonnell et al., 2008) covering Solar cycle 24 (2011–2019). Complementary measurements of solar wind and geomagnetic parameters were obtained from the NASA OMNIWeb database populated by NASA's Space Physics Data Facility with propagated values at the bowshock nose (https://omniweb.gsfc.nasa.gov/). These complementary data were used not only for the parameterization of the database but also for the statistical analysis performed in this study.

Figure 1 shows the steps followed in order to create the ULF Power Spectral Density (PSD) and $D_{LL}$ database, from the collection of the input data to the final data products. In detail, THEMIS magnetic and electric field vector data were pre-processed by transforming them into a Mean Field Aligned (MFA) coordinate system, similar to Balasis et al. (2013). The MFA is a local coordinate system defined by the ambient magnetic (or electric) field. The z axis is aligned with the unperturbed field (compressional or parallel component). The unperturbed field is obtained by a 30-min running average on the fields magnitude and, then, the compressional component is calculated as follows:

$$B_{com} = \Delta B \cdot \frac{B}{|B|} = \left( B - \bar{B} \right) \cdot \frac{B}{|B|} \tag{4}$$

where $\bar{B}$ is the unperturbed field. The y axis is perpendicular to the field's meridian pointing predominantly eastward (toroidal or azimuthal component), while the x axis completes the triad having an outward component (poloidal or radial component). After the transformation, the toroidal and compressional component of the electric and magnetic field, respectively, were de-trended using a 20-min moving average in order to eliminate the slow field variations. This is quite similar with a high-pass filtering with cut-off frequency at ≈0.83 mHz in order to focus on the Pc4-5 ULF frequencies.

The next step was to estimate the Power Spectral Density (PSD) of the waves in the 2 – 25 mHz frequency range for the two time-series, which corresponds to the drift-periods of near-equatorial mirroring electrons roughly in the 0.4-13 MeV. For the spectral analysis of the electric and magnetic field measurements we made use of the Continuous Wavelet Transform

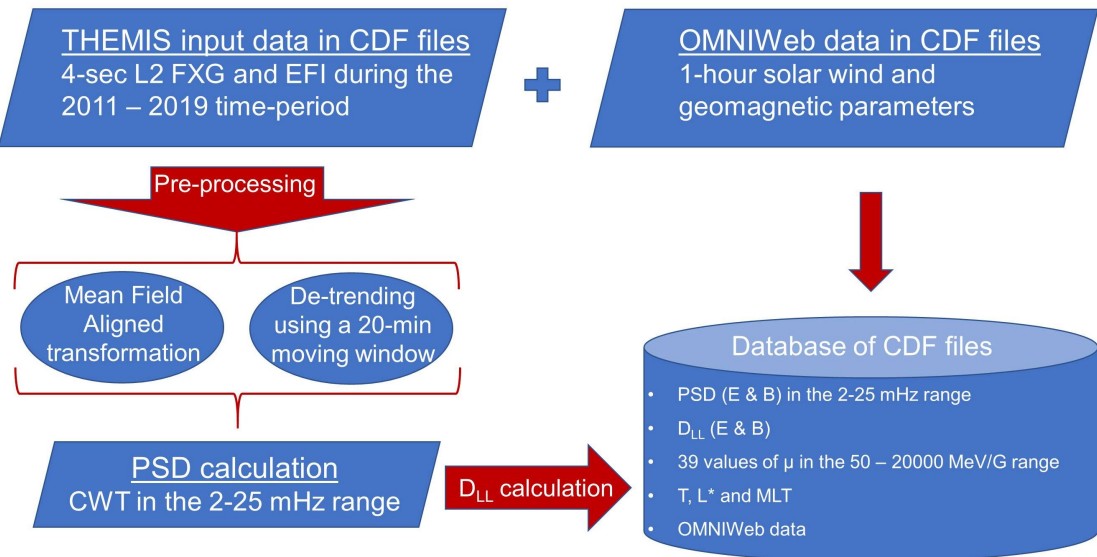

**Figure 1.** Work logic towards the creation of the SafeSpace radial diffusion coefficients database.

(CWT–see also Torrence and Compo (1998)) with the Morlet wavelet as the wavelet basis function (Morlet et al., 1983). Subsequently, the obtained 4-sec PSDs were averaged in 1 min windows, which were considered as one sample. For each time-stamp we also estimated the geomagnetic coordinates L* and Magnetic Local Time (MLT) using the International Radiation Belt Environment Modelling (IRBEM) library (Bourdarie and O'Brien, 2009) and the Olson–Pfitzer 1977 (Olson and Pfitzer,

1977) external magnetic field model. Note that, as the satellites move inbound and outbound with high velocities at low L-shells, the magnetic field measurements exhibit, not only orders of magnitude increase but, very large gradients as well. Therefore, after the application of the filtering, we perform a visual inspection of the time series and the steep gradients that have not been removed by 20-min averaging are removed by hand.

Finally, using the estimated PSDs, the $D_{LL}^B$ and $D_{LL}^E$ were calculated following the equations 2 and 3, respectively. For the

magnetic component, we have calculated $D_{LL}^B$ for 39 values of the first adiabatic invariant ($\mu$) in the 50–20000 MeV/G range.

The PSDs of both the toroidal electric and the compressional magnetic field as a function of time, L* and Magnetic Local Time (MLT), were stored in daily CDF files for each THEMIS probe, separately. Then, $D_{LL}^B$ and $D_{LL}^E$ were calculated as hourly values from the averaged PSD of all three THEMIS probes, and grouped in bins dL*=0.1 (the process and assumptions are discussed in detail in the following sections). These grouped radial diffusion coefficients were also stored in daily CDF

files. The database of both the ULF PSD and the $D_{LL}$ can be found at https://synergasia.uoa.gr/modules/document/?course= PHYS120.

## 2.2 Weighted average power

The wave power included in equations 2 and 3 corresponds to the power at a specific drift frequency for all m values, which essentially means that particles are radially transported via stochastic acceleration with various frequency waves (main frequency and harmonics). Nevertheless, to calculate the power at various m values, one would need at least 2m observations simultaneous in time, which is not trivial. To address this issue, it is often assumed that power at high m values is consistently lower than power at m = 1 and subsequently, that all power is contained in the lowest m = 1 wave mode of ULF waves driving diffusion (Ozeke et al., 2014). This assumption can lead to underestimation of the radial diffusion coefficient, since higher m values are shown to be often significant (e.g. m=2 up to m=5 at recovery phase of storms (see also Sarris et al. (2013)). To address this issue we have opted to use, in the place of power at a specific frequency, the weighted average power over the whole frequency range under study (in our case Pc4 and Pc5 frequency range). This weighted average power is given by Torrence and Compo (1998) as follows:

$$P_{total} = \left( \frac{dj \cdot dt}{Cdelta} \right) \cdot \sum_{f} PSD(f) \tag{5}$$

where Cdelta is a smoothing factor (which for the Morlet wavelet is empirically derived as 0.776) and $dj = -\frac{log_2 \left( \frac{f_{min}}{f_{max}} \right)}{1/f_{min}}$ is the sampling scale (for more details see also section 3 of Torrence and Compo (1998) and section 4 of Katsavrias et al. (2022)).

## 2.3 Assumptions

Even though we have followed a well-established methodology in order to calculate–as accurately as possible–the ULF PSD and the corresponding $D_{LL}$ there are still worth-mentioning assumptions, which are based on the theoretical approach we have used as well as on the inherent limitations of the in-situ data.

As already discussed, important differences can exist between the two approaches by Fei et al. (2006) and by Falthammar (1965). It is estimated that the former can underestimate (compared to the latter) the total $D_{LL}$ by a factor of 2 (Lejosne, 2019), given that the magnetic field disturbances are described by the simple model introduced by Falthammar (1965) and that there is no electric potential disturbance. Nevertheless, Fei's approach is the more widely used due to the fact that it is very difficult to separate the total measured electric field from single point measurements in space (Brautigam et al. , 2005) into its convective and inductive components (Lejosne and Kollmann , 2020). Furthermore, it has been shown that this factor-of-2 discrepancy is comparatively minor relative to the large variability in the observed values (Sandhu et al., 2021).

In addition, the theoretical approach of Fei et al. (2006) formulas apply for equatorially mirroring particles only, while THEMIS satellites do not necessarily sample the magnetic equator. Nevertheless, they remain very close to the magnetic equator throughout their trajectories in the heart of the outer belt (Angelopoulos, 2008; Turner et al., 2012b) something that allows us to assume that the uncertainty in the $D_{LL}$ calculation will be rather small. Moreover, the distribution of both the electric and magnetic field power with respect to the Bratio ($B_{eq}/B_{local}$, not shown here) shows that the uncertainty in ULF power is up to a factor of 2, at least for Bratio values larger than 0.8. We emphasize that this 0.8 threshold in Bratio is applied

in our database in order to minimize the corresponding uncertainty, thus all the following results correspond to $D_{LL}$ values at points with Bratio>0.8. We also note that it is difficult to differentiate between spatial and temporal field variations. In our calculations we have assumed that since the spatial variations, especially at the radial distances corresponding to the outer belt, are usually slow variations, they are filtered out by the filtering process. This is of course usually not possible at very low L-shells, thus the corresponding calculations are removed from the dataset.

Finally, we have to mention the uncertainties introduced by the use of the Olson-Pfitzer quiet model for the estimation of the magnetic ephemeris data, as well as the uncertainties of the instruments used (e.g. Califf et al. (2015)).

## 3   Uncertainties generated by the azimuthal dependence of ULF power

Equations 2 and 3 implicitly assume a uniform distribution of wave power in azimuth. In reality, the azimuthal distribution of the wave power in the Pc4-5 range depends on their generation mechanism, e.g. the wave power due to the Kelvin-Helmholtz instability is expected to be greater near dawn and dusk sectors, while due to the pressure pulses from the solar wind is expected to be greater near noon. Furthermore, radial diffusion coefficients are by definition drift-averaged which is opposite in paradigm to these existing physical anisotropies. However, the in-situ calculation of radial diffusion coefficients with full azimuthal coverage, and therefore averaging, would require a large spacecraft constellation with appropriate positioning providing concurrent and intensity coherent measurements. Obviously this is not currently possible. Our efforts have been focused here on quantifying the magnitude of radial diffusion due to ULF waves observed by the THEMIS spacecraft, which provide a maximum MLT coverage that rarely exceeds 6 hours per hour and per L*.

In order to discuss the possible uncertainties generated by the limited azimuthal coverage we calculate an 1-min resolution proxy of the $D_{LL}$ at each point of the spacecraft orbit, for each spacecraft separately. Since, the $D_{LL}$ proxy, at each L* value, has been calculated as the product of the weighted averaged power with a simple multiplication factor it is expected to reflect directly the azimuthal distribution of wave power for both the magnetic and the electric component. Figure 2 shows the logarithms of the mean $D_{LL}$ proxy as a function of MLT and L* for three levels of geomagnetic activity: Kp< 3 (left column panels), 3 <Kp< 5 (middle column panels) and Kp> 5 (right column panels).

As shown, there are obvious differences between the two components. During quiet times, the $D_{LL}^E$ proxy (top left panel) exceeds the value of 10 days$^{-1}$ outside the geosynchronous orbit and is approximately equal to 1 days$^{-1}$ at the 4.5–6 L* range, while there is a significant MLT asymmetry at the 0–3 MLT range. As we move to higher geomagnetic activity levels (3 <Kp< 5–top middle panel), $D_{LL}^E$ proxy intensifies and, in addition, this asymmetry becomes stronger and reaches even lower L* values. During intense geomagnetic activity levels (top right panel), $D_{LL}^E$ proxy values range between 10 and 100 days$^{-1}$ at L*> 5 and they reach approximately the value of 1 days$^{-1}$ even down to at L*= 3.5. On the other hand, the $D_{LL}^B$ proxy exhibits a different behaviour. During quiet times (bottom left panel), the $D_{LL}^B$ proxy values reach 1 days$^{-1}$ at L*> 7 and only at the dayside sector (approximately in the 9–15 MLT range). As we move to higher geomagnetic activity levels, the $D_{LL}^B$ proxy exceeds the value of 10 days$^{-1}$ even inside the geosynchronous orbit L*< 6. Furthermore, the MLT asymmetry becomes more intense and wide (approximately in the 5–18 MLT range during Kp> 5 periods).

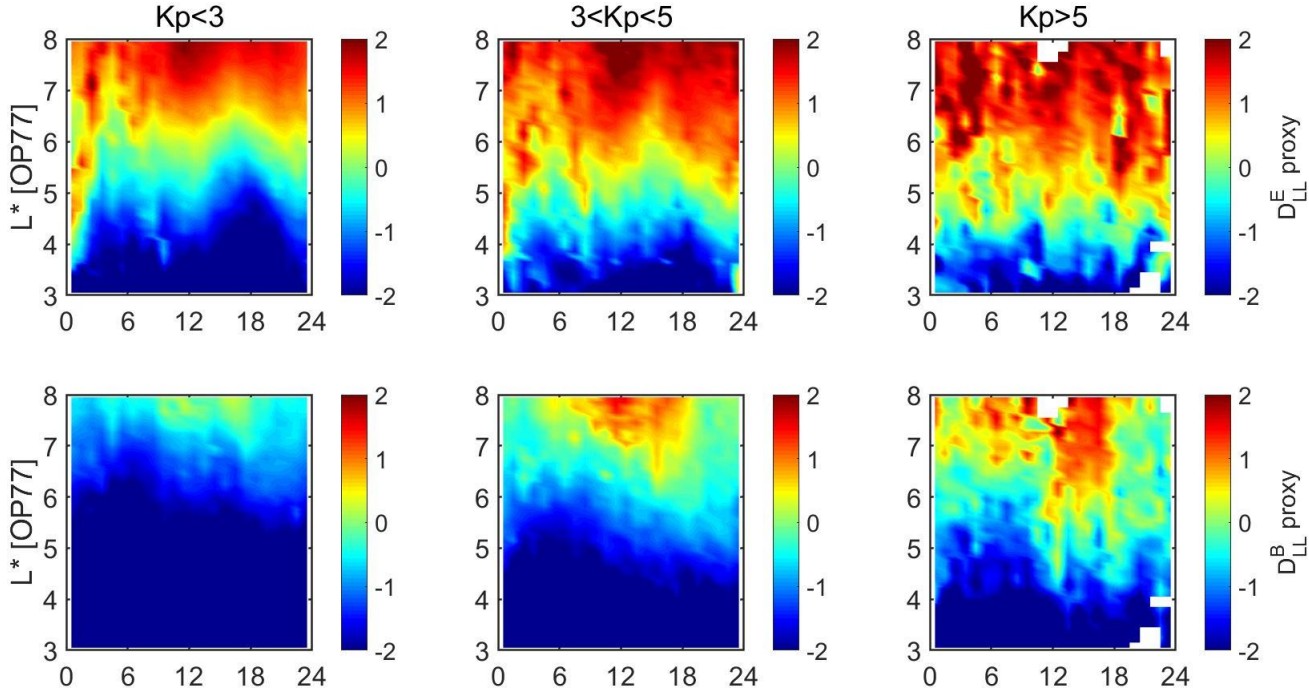

**Figure 2.** Logarithms of the mean $D_{LL}$ proxy as a function of MLT (dMLT=1 hour) and L* (dL*=0.1) for three levels of geomagnetic activity: (left column panels) Kp< 3, (middle column panels) 3 <Kp< 5 and (right column panels) Kp> 5. Top and bottom row panels correspond to the electric and the magnetic component (for $\mu$=1000 MeV/G) of the $D_{LL}$ proxy, respectively.

The aforementioned feature of the $D_{LL}^B$ proxy is in agreement with the correlation results shown in figure 3 and indicates that the magnetic field, and consequently the magnetic component of the diffusion coefficient, is linked with ULF waves generated through solar wind pressure pulses (Kepko et al., 2002). On the other hand, the observed asymmetry in the $D_{LL}^E$ proxy is not only linked with solar wind speed but with internal mechanisms such as substorm activity, especially during quiet

or moderate magnetospheric activity. This is supported by the remarkable agreement of the $D_{LL}^E$ proxy with the findings of Nosé et al. (1998), who showed that substorms generate azimuthal ULF fluctuations at the nightside which peak at 1–2 MLT. Furthermore, this is also in agreement with the results of figure 3 and the significant correlation of $D_{LL}^E$ with the AE index. These results are also in good agreement with Sandhu et al. (2021) who used Van Allen probes data to infer the radial diffusion coefficients. This agreement indicates that the uncertainty introduced by the magnetic latitude (and already discussed in section

2) is insignificant, even though there is no straightforward comparison with the dataset used by the latter authors.

Our $D_{LL}$ calculations employ a fraction of the full azimuthal coverage and, thus, are expected to have some uncertainty. As shown in figure 2, the assymetry in the electric component of the $D_{LL}$ proxy is limited in the 0-3 MLT range for low to moderate geomagnetic conditions, which means that the uncertainty in the $D_{LL}^E$ will not be significant given the up to 6 hours coverage in MLT from the three THEMIS spacecraft. In addition, the magnetic component of the $D_{LL}$ proxy during low

geomagnetic activity does not exhibit significant variation with the MLT, which also corresponds to insignificant uncertainties

in the $D_{LL}^B$. On the other hand, the $D_{LL}^B$ proxy during moderate and intense geomagnetic activity exhibits an approximately one order of magnitude difference between dayside and nightside. This means that the partial azimuthal coverage provided by the three THEMIS spacecraft could lead to an up to one order of magnitude of uncertainty in the estimation of the $D_{LL}^B$ for particular spatial configurations of the three THEMIS spacecrafts, e.g. when all three are located in the nightside or all three are located in the dayside. We note that such uncertainties are present in all $D_{LL}$ time-series estimated by in-situ measurements, e.g. Jaynes et al. (2018); Olifer et al. (2019); Sandhu et al. (2021) as well as for semi-empirical models, e.g. Ozeke et al. (2014) have used only dayside measurements from ground-based magnetometers to infer the electric component of the $D_{LL}$.

## 4    Dependence on solar wind and geomagnetic parameters

Figure 3 shows the Pearson correlation coefficients (henceforward CCs) between the logarithm of the hourly mean values of $D_{LL}$ with various solar wind parameters, geomagnetic indices and coupling functions in the 3–8 L* range. Shown, from left to right, are the Interplanetary Magnetic Field (IMF), solar wind velocity, pressure and number density, plasma $\beta$ parameter, the geomagnetic indices SYM-H, AE and Kp, the $\epsilon$ parameter (Akasofu, 1981), the southward solar wind field (here we show the exponential of Bs), the Half-Wave Rectifier (Burton et al., 1975) and Newell's function (Newell et al., 2007).

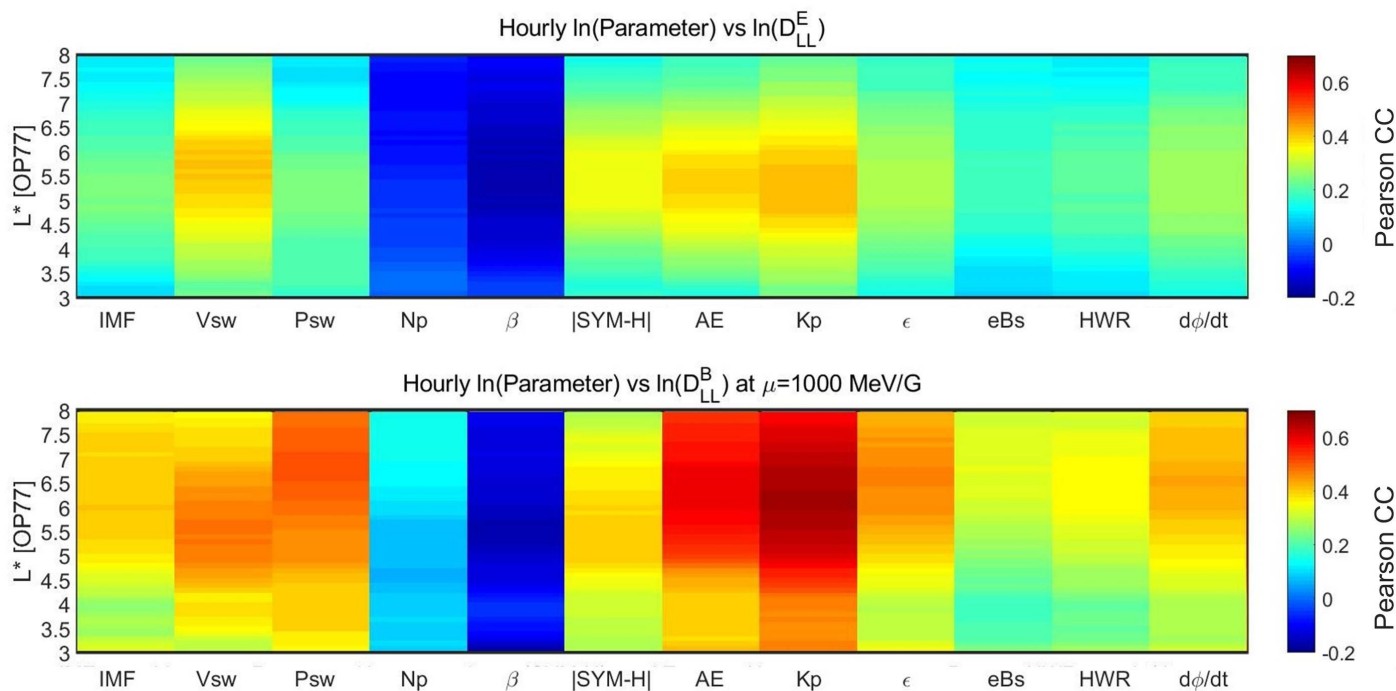

**Figure 3.** Pearson correlation coefficients between the natural logarithms of the hourly mean values of $D_{LL}^E$ (top panel) and $D_{LL}^B$ (for $\mu = 1000$ MeV/G–bottom panel) with various solar wind parameters, geomagnetic indices and coupling functions as a function of L* (with dL*=0.1).

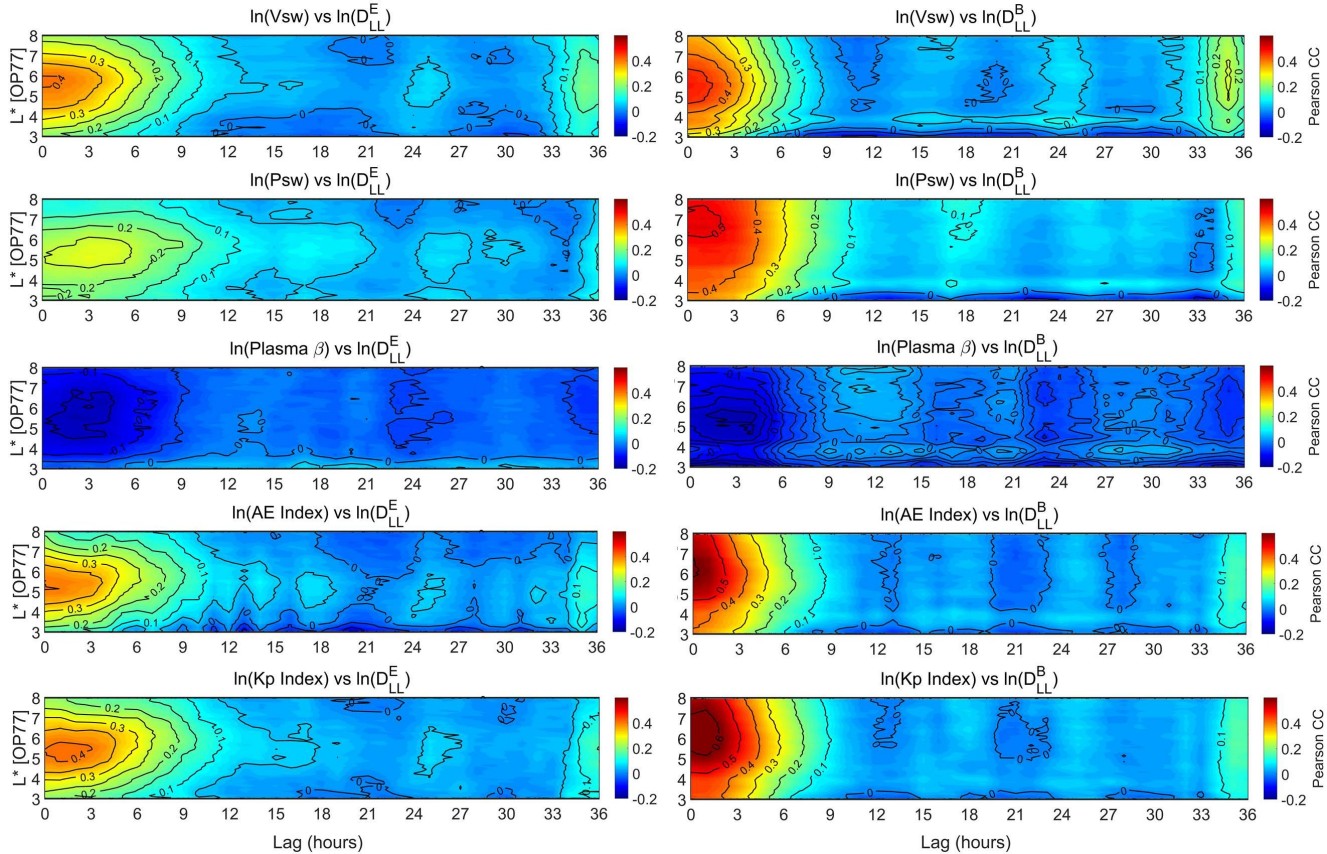

**Figure 4.** Pearson correlation coefficients between the natural logarithms of the hourly mean values of $D_{LL}^E$ (left panels) and $D_{LL}^B$ (for $\mu = 1000$ MeV/G–right panels) with (top to bottom) solar wind speed, dynamic pressure, plasma $\beta$, AE and Kp index. CCs are presented–both color coded and with the black contours–as a function of L* (with dL*=0.1) and the time-lag from 0 to 36 hours.

Generally, the CCs of the magnetic component exhibit greater values than the ones of the electric component with maxima at $\approx 0.7$ and $\approx 0.4$, respectively. Note that we only show the $D_{LL}^B$ at 1000 MeV/G but the CCs do not change at all if we account for the $\mu$ value. In detail, both $D_{LL}$ components exhibit their best correlation with the geomagnetic indices AE and Kp. Nevertheless there is a pronounced difference concerning the L* location of the maximum CC. For the electric component the maximum CC ($\approx 0.4$ for both AE and Kp) is located roughly at the 4.5–6.5 L* range. For the magnetic component, the maximum CC with AE ($\approx 0.65$) is located roughly at the 4.5–8 L* range and the maximum CC with Kp ($\approx 0.7$) covers approximately the whole L* range. The latter is in agreement with Dimitrakoudis et al. (2015) who found that the Kp index provides the best parameterization of the $D_{LL}^B$. Our results indicate that this parameterization may not work equally for the electric component, especially for L* values higher than 6.5 and lower than 4.5.

Furthermore, the CC between solar wind speed and $D_{LL}$ is at $\approx 0.4$ and $\approx 0.5$ for the electric and magnetic component, respectively, but both at the 4.5–6.5 L* range. The importance of magnetopause instabilities–induced by the increased solar

wind velocity–has been well established before (Bentley et al., 2018) but here we show that it can similarly affect both $D_{LL}$ components. Another interesting feature is exhibited by the correlation between the $D_{LL}$ and solar wind dynamic pressure even though there is no significant correlation with number density. For the electric component the CC does not exceed the 0.2 value but for the magnetic component is larger than 0.5 at L*> 4.5. A possible explanation of this feature could be that, since solar wind pressure pulses produce mainly global magnetospheric oscillations (Kepko et al., 2002; Takahashi et al., 2012), they do not affect the azimuthal electric field variations and thus the electric $D_{LL}$ component.

It is worth mentioning that the only parameter which exhibits an anti-correlation with the $D_{LL}$ is the plasma $\beta$ parameter at all L* values. Nevertheless, the maximum CC at both components does not exceed -0.2. Finally, the CCs between the $D_{LL}^B$ component with Newell's function and Akasofu's $\epsilon$ parameter exhibit a similar trend with AE index but with lower CC maxima ($\approx$0.4). This is expected since these parameters are known to be well correlated with substorm activity (Katsavrias et al., 2021).

Figure 4 shows the cross-correlation between $D_{LL}^E$ (left panels) and $D_{LL}^B$ (for $\mu = 1000$ MeV/G–right panels) with (top to bottom) solar wind speed, dynamic pressure, plasma $\beta$, AE and Kp index. Note that in this figure we are showing only the parameters which, according to figure 3, exhibited noteworthy correlations. Similar to figure 3, the CCs of the magnetic component are systematically higher than the ones of the electric component, at least for time-lags up to 12 hours, with the exception of plasma $\beta$. As shown, the maximum CCs for the magnetic component (right panels) are exhibited at zero time-lag, while they become negligible for time-lags greater than 9 hours. A similar trend is exhibited for the CCs of the electric component with solar wind speed and AE index. On the contrary, the CC of the electric component with Kp index exhibits a maximum at the 0–3 hours time-lag.

## 5 ICME vs SIR driven geospace disturbances

The role of solar wind drivers (e.g. Interplanetary Coronal Mass Ejections–ICMEs and Stream Interaction Regions–SIRs) has been suggested to play an important role to the generation of ULF waves and, consqently, to the evolution of radial diffusion coefficients (Simms et al., 2010; Kilpua et al., 2015). In order to investigate the dependence of the $D_{LL}^E$ and $D_{LL}^B$ on the solar wind driver we have selected 25 ICME– and 46 SIR–driven geospace disturbances (71 events in total) in the 2011–2019 time period, following the criteria of Katsavrias et al. (2019b). More specifically, we have chosen events that include a single driver and have no pre-conditioning in solar wind parameters for at least 12 hours before the arrival of the ICME or SIR. Since we have applied no criteria depending on the Dst index (non-storm events are also included), we have used as zero-epoch time ($t_0$) the time of the maximum compression of the magnetopause ($Lmp_{min}$) as it is given by the empirical model of Shue et al. (1998).

Figure 5 shows the results of the superposed epoch analysis. As shown, both groups exhibit several differences. During ICME driven disturbances the maximum increase in $D_{LL}^E$ takes place on $t_0$ at all L*> 4 and reaches a median value of 1000 days$^{-1}$ at L*> 5, while significant activity reaches down to L$\approx$ 3.5 up to 12 hours. After these 12 hours and the activity is still significant at L*> 5 and lasts up to 96 hours (4 days). During SIR driven disturbances, the $D_{LL}^E$ exhibits a quite similar trend (it lasts up to 4 days after $t_0$) but both its magnitude and the penetration to inner L* are lower compared to the ICME driven

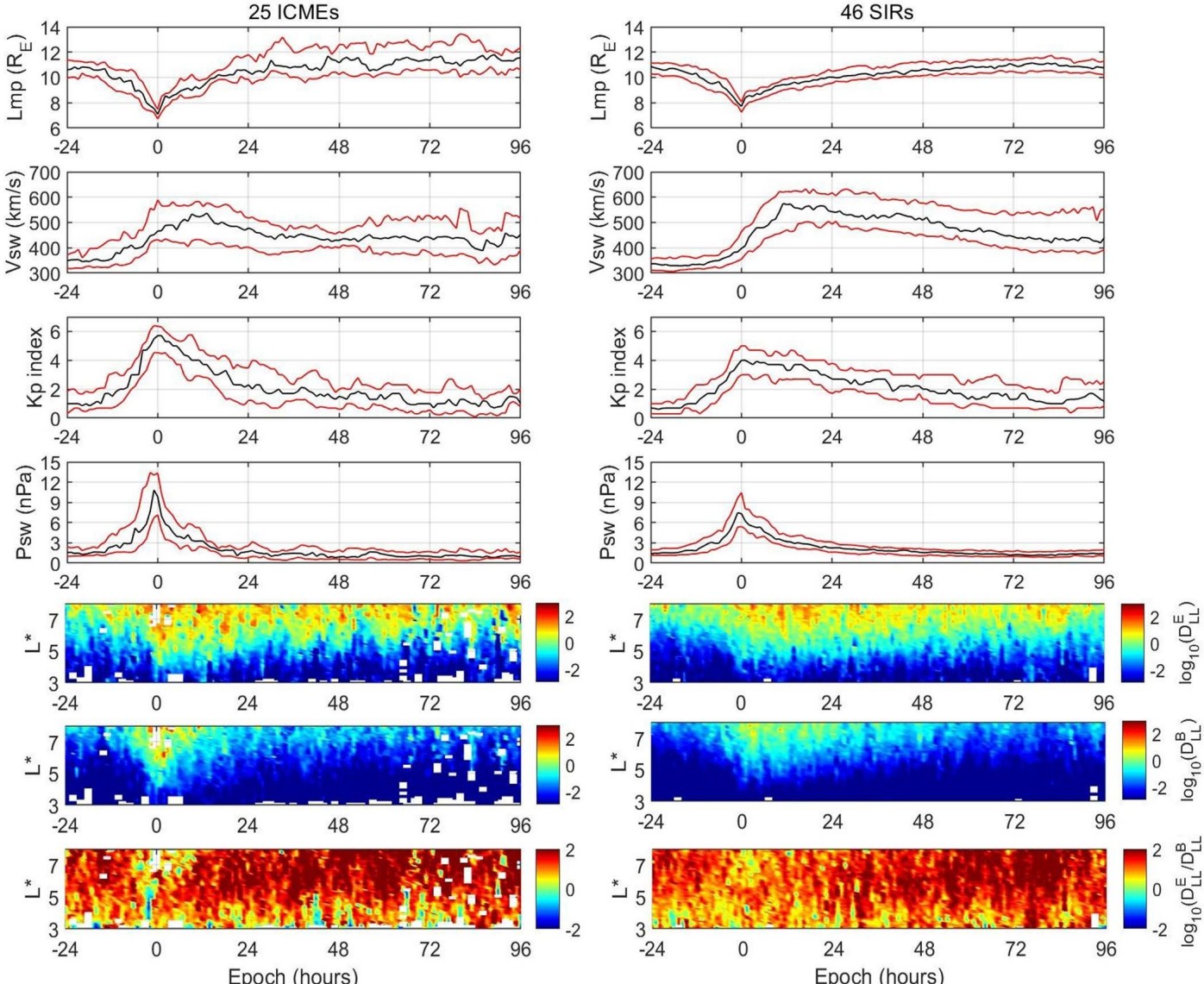

**Figure 5.** Superposed epoch analysis of the 25 ICME (left column panels) and 46 SIR (right column panels) driven geospace disturbances. Top to bottom: median (black line), $25^{th}$ and $75^{th}$ quantiles (red lines) of the magnetopause location predicted by Shue et al. (1998) model, solar wind speed, Kp index, solar wind dynamic pressure, the logarithm of the median values of $D_{LL}^E$, $D_{LL}^B$ (for $\mu$=1000 MeV/G) and their ratio. The binning is performed with dt=1 hour and dL*=0.1.

disturbances. On the other hand, the $D_{LL}^B$ exhibits much more pronounced differences. During ICME driven disturbances the maximum increase in $D_{LL}^B$ takes place on $t_0$ and the penetration of the activity reaches down to L*≈4. The overall enhancement occurs on -8< $t_0$ <12 hours. During SIR driven disturbances, the $D_{LL}^B$ hardly reaches L*≈4 and the maximum increase reaches

a value of 10. Nevertheless, the overall activity lasts up to approximately 30 hours after $t_0$. Furthermore, the enhancement as well as the penetration of $D_{LL}^B$ to low L*, is very well correlated with the enhancement in both solar wind dynamic pressure and Kp index and, consequently, is in agreement with the findings of figure 3. This result is also in agreement with Simms et al. (2010) who indicated that ground Pc5 power was greater during CME storms, especially during the main and recovery phase. One step further, Kalliokoski et al. (2020) studied 37 ICME-driven sheath regions in the Van Allen Probes era and linked the increased Pc5-ULF activity at GEO with the increased pressure during the sheath.

Finally, a very important feature is exhibited by the ratio of the electric over the magnetic component, which generally spans the 0.1–100 range. As shown in the bottom panels of figure 5, the electric component is mostly dominant–up to two orders of magnitude compared with the magnetic component. This feature changes dramatically during ICME driven disturbances and around $\pm$ 6 hours from the maximum compression of the magnetopause where the $D_{LL}$ ratio decreases below 1 at all L* values. Furthermore, at L*> 6, the $D_{LL}$ ratio is approximately 1 up to 12 hours after $t_0$. The relative strength of the two $D_{LL}$ has been discussed before by Olifer et al. (2019) who studied the components ratio during the St. Patricks event of 2015. These authors indicated that during the main phase of this ICME driven storm, the magnetic component exceeded the electric by approximately one order of magnitude, something that semi-empirical models cannot reproduce. Here we replicate this result using a statistical sample of 25 ICME driven disturbances independent of the magnitude of Dst index. Note that this feature, even though it is not that obvious, may be important during SIR disturbances as well. As shown in the bottom right panel of figure 5, the $D_{LL}$ ratio at L*> 5.5 is decreased from approximately 100 to approximately 1 at $\pm3$ hours from $t_0$. We suggest that this difference in the $D_{LL}$ ratio between ICME and SIR–driven disturbances is probably attributed to the existence (or not) of shocks, which produce significant increase of the dynamic pressure and accompany, more often, the ICME–driven events.

We must emphasize the fact that this feature introduces a significant energy dependence on the $D_{LL}$, since the magnetic component is energy dependent, that may be of great importance to radiation belt simulations. Furthermore, this feature is expected to be dependent on the first adiabatic invariant as well, since greater values of $\mu$ produce greater values of $D_{LL}^B$, which will consequently lead to changes in $D_{LL}$ ratio. It is also expected that, except the magnitude, the change in $\mu$ will affect both the duration and the L* coverage of this feature as well. In a future study we intend to investigate in greater detail these changes.

## 6 The use of $D_{LL}$ time-series in physics-based models

### 6.1 Comparison with semi-empirical models

As already discussed in the introduction section, even though the semi-empirical Kp-parameterized models have the advantage of providing estimations/predictions of the $D_{LL}$ without the dependence on the in-situ measurements, they can significantly deviate from the calculated $D_{LL}$ time-series. In order to statistically establish these deviations we directly compare the calculated $D_{LL}$ values ($D_{LL}^B$ is always at $\mu$=1000 MeV/G) from the SafeSpace database to the empirically modelled values of table 1 for the whole 2011–2019 time period.

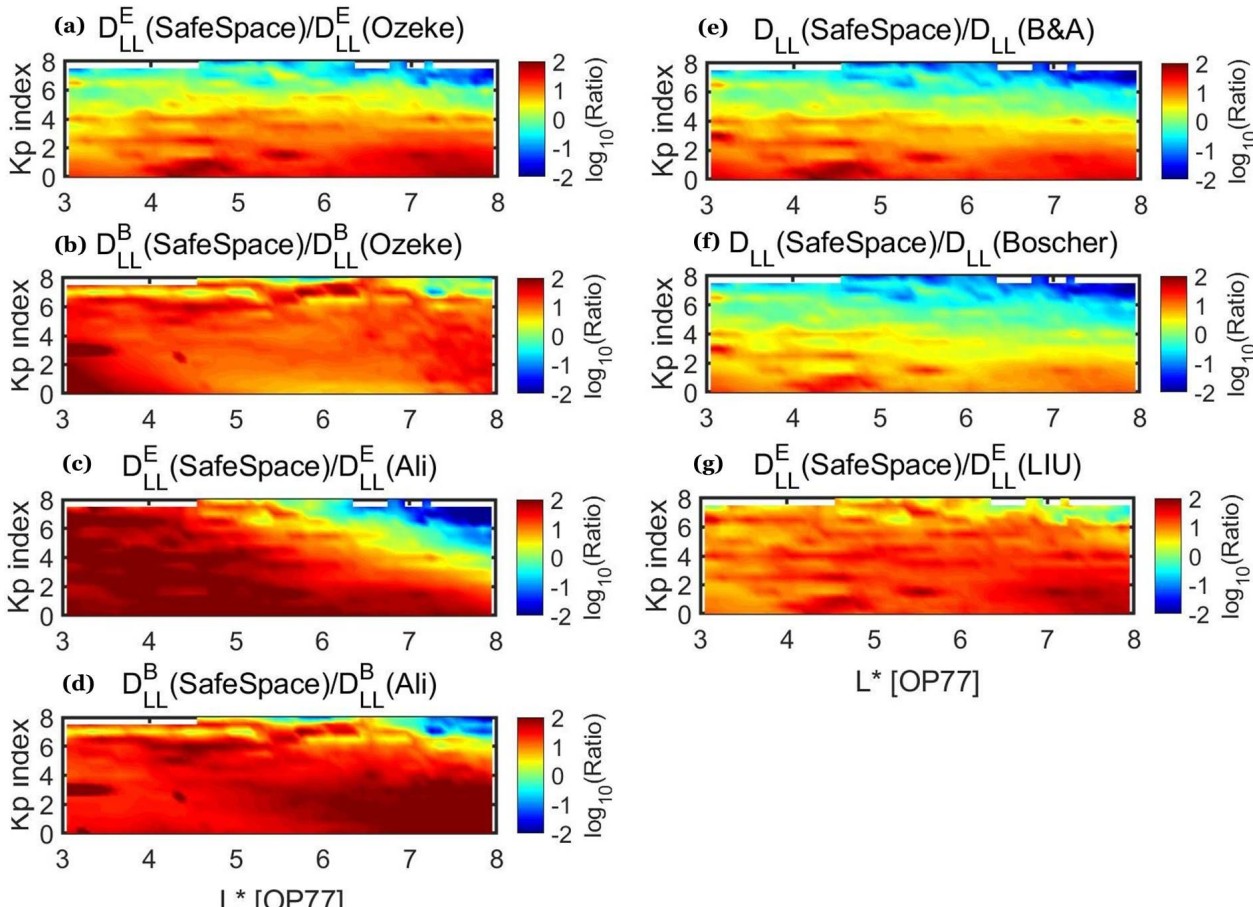

**Figure 6.** Comparison of the SafeSpace $D_{LL}$ values with the 5 semi-empirical models listed in table 1 binned in Kp index values (dKp=0.5) and L* (dL*=0.1). $D_{LL}^B$ corresponds to $\mu$=1000 MeV/G.

We note that the comparison of radial diffusion coefficients among multiple methods is anything but a straightforward process since the details of each method are different (e.g. different datasets, different time-periods in a solar cycle or different theoretical approaches). In addition, here we attempt a comparison of a dataset inferred from in situ measurements with the estimations of semi-empirical models, which are by definition two different things. Nevertheless, we attempt a comparison in order to show a clearer picture of the statistical behaviour of each model compared to our $D_{LL}$ inferred from in-situ data.

Figure 6 shows this comparison parameterized by Kp index. As shown, there is a general trend with all empirical models (and their components) where the $D_{LL}$ has lower values at low levels of geomagnetic activity at all L* and higher values at high levels of geomagnetic activity at high L* values, compared to the SafeSpace $D_{LL}$. In detail, concerning the Ozeke et al. (2014) model, the $D_{LL}^E$ exhibits lower values at all L* for Kp< 4 (panel a), while there is a relatively good agreement with SafeSpace $D_{LL}$ at 4 <Kp< 6. The observed higher values at L*> 7 and Kp> 6 cannot be discussed since it exceeds the limits

of the Ozeke model (see also table 1). These features are in good agreement with the results of Sandhu et al. (2021) and Murphy et al. (2015) even though the latter authors performed a statistical comparison during storm time only and with $D_{LL}$ values calculated using RBSP and ground-based data, respectively. On the other hand, the $D_{LL}^B[OZ]$ (panel b) exhibits persistently lower values of at least a factor of 10 at all L* and Kp values, which is in agreement with the results of Olifer et al. (2019). Also note that this feature is expected to be even more pronounced with increasing $\mu$ values, since the $D_{LL}^B[OZ]$ is not energy dependent. A possible source for the aforementioned disagreement between the Ozeke model and our calculated $D_{LL}$ may come from the assumption that ULF wave power is concentrated in the lowest mode (m = 1) thus underestimating the true ULF wave power, while in this study, a weighted averaging over all frequencies has been used. Furthermore, the $D_{LL}^E[OZ]$ is based on ground magnetometer measurements restricted in the dayside sector with the assumption that the observed power is independent of the MLT. This is in contrast with our results in figure 2 where the contribution from internal mechanisms (i.e substorm activity) are shown to be important. The most striking difference is the persistent disagreement of the $D_{LL}^B[OZ]$ even though it is also derived by THEMIS magnetic field measurements. We suggest that the primary sources of this feature are the m=1 assumption and the fact that its training dataset is restricted to the 2007-2011 time period, thus including mostly quiet magnetospheric conditions.

It is worth-mentioning that the Brautigam and Albert (2000) model (panel e) exhibits very similar trend with the electric component of the Ozeke et al. (2014) model even though the two models have been developed using completely different theoretical approaches and datasets.

Concerning the Ali et al. (2016) model, both components exhibit a significantly lower values compared to the SafeSpace $D_{LL}$ that reaches approximately two orders of magnitude (panels c and d), with the exception of $D_{LL}^E[ALI]$ which appears higher at high L* and Kp values (top right corner of panel c). Nevertheless, this area is outside the limits of the model as described in table 1. The overall behaviour of the Ali et al. (2016) model presented in this figure is in agreement with the results of Drozdov et al. (2021) who showed that simulations performed with the Versatile Electron Radiation Belt (VERB) code using this $D_{LL}$ model exhibited significantly lower flux levels. Similarly with Ozeke et al. (2014), the $D_{LL}[ALI]$ are calculated with the assumption that the observed power is independent of the MLT and that all power falls into m=1. Moreover, the Ali et al. (2016) model considers ULF wave power in a narrow frequency range (1.67–6.67 mHz), while in this study we have considered the full Pc4-5 range up to 25 mHz. We suggest that the aforementioned differences possibly account for the lower values of the $D_{LL}[ALI]$ in comparison with our calculated ones.

At the same extent, the Liu et al. (2016) model for the $D_{LL}^E$ (panel g) exhibits mostly lower values, compared to our calculated ones, up to a factor of 10, even though it also uses THEMIS electric field measurements to derive the electric component of the $D_{LL}$. In addition, the Liu model exploits the entire Pc3-5 frequency range to derive the $D_{LL}^E$, which is quite similar to the range used in this study, but the same assumption that all power falls into m=1 is used here as well. Nevertheless, a significant difference is that the dataset used to derive the model spans the 2008-2014 time period. This time period includes the extended minimum of 2008-2009 (which is not included in our dataset) and misses the recovery phase of Solar cycle 24 where several intense events occurred.

The Boscher et al. (2018) model (panel f) exhibits very similar results compared to the SafeSpace $D_{LL}$. In detail, the modelled $D_{LL}$ is in good agreement for $4 <Kp< 7$ at $L*< 6$ and for $3 <Kp< 6$ at $L*> 6$. Nevertheless, it exhibits significantly lower values of the $D_{LL}$ up to a factor of 10 during quiet times at all $L*$ and significantly higher values (at least a factor of 10) for $Kp> 6$ approximately outside the geosynchronous orbit, which nevertheless is outside the limits of the model.

Finally, we have to consider the uncertainties in the SafeSpace calculated $D_{LL}$ that may also be a cause of disagreement versus the aforementioned semi-empirical models. As discussed in section 2, it has been shown that the Fei et al. (2006) approach can underestimate the radial diffusion coefficient by a factor of two compared with the Falthammar (1965) approach. This is sufficient to explain the difference exhibited by SafeSpace $D_{LL}$ and the Brautigam and Albert (2000) and Boscher et al. (2018) models at $L*> 4$, but it cannot explain the up to a factor of 10 difference at lower L-shells. Another uncertainty, also discussed in section 2, comes from the limited azimuthal coverage of THEMIS satellites used in this study. Nevertheless, the results of figure 6 are averaged values of the SafeSpace $D_{LL}$ for specific values of $L*$ and Kp over a 9 years time-period, thus including all MLT values. Last but not least, we have to consider the uncertainties introduced by the use of the OP77 model, especially at high L-shells (outside GEO) and high Kp values.

## 6.2   St. Patrick's 2015 event

In the previous section we presented an extended comparison of the various semi-empirical models with the calculated $D_{LL}$ from the SafeSpace database showing that all of them (more or less) exhibit significant deviations at different $L*$ and Kp values range. These deviations correspond to the cause ($D_{LL}$) and not the effect (electron radial diffusion). In order to evaluate the actual effect of these calculated radial diffusion coefficients on the outer belt dynamics we have performed simulations without the energy diffusion term using the Salammbô model. Figure 7 shows the results of this simulation for two electron energies at 500 (left column panels) and 1500 keV (right column panels) during the March 2015 time period which includes the St. Patrick's event of March 17. Note that the magnetospheric model used in the simulations is the Olson-Pfitzer quiet model. In addition, the dataset gaps due to the partial azimuthal coverage of THEMIS constellation were filled using power law interpolation/extrapolation schemes in $L*$, and then a time interpolation at each $L*$ value was performed.

As shown in the 500 keV electron energy, simulation results exhibit more intense radial transport at the outer edge of the outer radiation belt ($4<L*<5.5$) both during the relatively quiet period on early March and during the intense St. Patricks storm when using the calculated $D_{LL}$ compared to the Boscher et al. (2018) model. This is in agreement with the results shown in figure 6 where the semi-empirical models underestimate the $D_{LL}$ at high $L*$ values during active geomagnetic conditions. Moreover, as shown in the 1500 keV electron energy, the simulation captures more realistically, not only the re-distribution of the relativistic electron population, but generally the dynamics and the magnitude of the 1500 keV electron fluxes. The latter is particularly important since it has been reported that during the St. Patricks event of 2015, radial diffusion contributed not only to the enhancement of 1-2 MeV electrons but also to further acceleration to ultra-relativistic energies (Jaynes et al., 2018). We note that the magnitude of the flux in the Salammbô simulation is not expected to agree with the MagEIS data due to the lack of the energy diffusion term (in-situ acceleration by VLF chorus waves), which for the St. Patrick's event of 2015 has been shown to be crucial especially for 1-2 MeV electrons (Li et al., 2016).

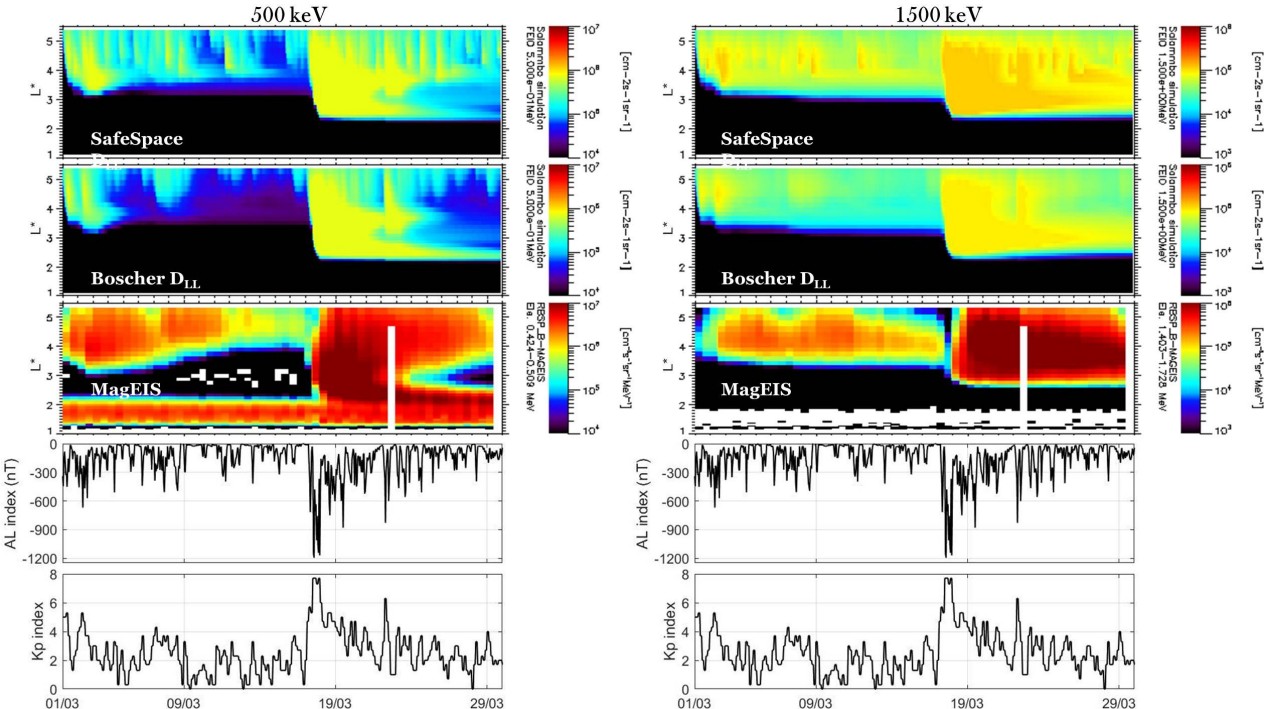

**Figure 7.** Simulation of the outer radiation belt dynamics during the St. Patrick's event of 2015 using the Salammbô model for two electron energies: (left column panels) 500 keV and (right column panels) 1.5 MeV. From bottom to top are shown the Kp index, the AL index, the electron flux measured by the MagEIS instrument on board the Van Allen probes, the simulation results using the Boscher et al. (2018) semi-empirical model and the simulation results using the SafeSpace $D_{LL}$ values.

We must emphasize the fact that the aforementioned comparison is performed between the calculated $\mu$-dependent $D_{LL}$ from the SafeSpace database and the Boscher et al. (2018) model, only. This is done in accordance to the results discussed in the previous section (see also figure 6) where we showed that the Boscher model exhibited the best comparison with the SafeSpace calculated diffusion coefficients. It is also notable that the simulations results indicate that the uncertainties introduced by the limited azimuthal coverage provided by the THEMIS spacecraft are rather small compared to the uncertainties of using medians or other statistical quantities over long-term datasets.

## 7 Conclusions

In the framework of the SafeSpace project we have used 9 years (2011 – 2019) of multi-point magnetic and electric field measurements from THEMIS A, D and E satellites to create a database of ULF power spectral density and the estimated radial diffusion coefficients. We have further exploited this database in order to investigate the dependence of these calculated $D_{LL}$ on several solar wind and geomagnetic parameters, to solar wind drivers (ICMEs and SIRs), with respect to L*.

The results of this analysis can be summarized as follows:

1. $D_{LL}^B$ exhibits higher correlation coefficients, with both solar wind and geomagnetic parameters, compared to $D_{LL}^E$. In addition, $D_{LL}^E$ exhibits its highest coefficients at the 4.5-6.5 L* range and $D_{LL}^B$ at L*>4.5.

2. Both $D_{LL}$ components (magnetic and electric) exhibit good correlation with Kp and AE index. Furthermore, $D_{LL}^E$ exhibits good correlation with solar wind speed, while $D_{LL}^B$ exhibits good correlation with both solar wind speed and pressure with zero time-lag.

3. The superposed epoch analysis reveals important differences between the evolution of $D_{LL}$ during ICME- and SIR-driven disturbances. During the former, the high solar wind pressure values combined with the intense magnetospheric compression produce $D_{LL}^B$ values comparable or even greater than the ones of $D_{LL}^E$. This feature cannot be captured by semi-empirical models and introduces a significant energy dependence on the $D_{LL}$.

Furthermore, the comparison of the semi-empirical models with the $D_{LL}$ from the SafeSpace database reveals significant deviations depending on the level of geomagnetic activity and the drift shell. Generally, all models exhibit lower values of the $D_{LL}$ during quiet times at low L* values, and higher values during high levels of geomagnetic activity at high L* values, compared to the SafeSpace $D_{LL}$. Finally, we have evaluated these calculated $D_{LL}$ through simulations of relativistic electrons using the Salammbô code. The simulations show that the model captures more realistically, not only the re-distribution of the relativistic electron population, but generally the dynamics and the magnitude of the electron fluxes when using the SafeSpace $D_{LL}$ compared with the well-established Boscher semi-empirical model.

We believe that these results may offer significant insight for future modelling efforts in order to develop an accurate now-casting/forecasting model for radial diffusion coefficients.

*Data availability.* The scientific products of the SafeSpace radial diffusion coefficients database can be found at https://synergasia.uoa.gr/modules/document/?course=PHYS120.

*Author contributions.* CK drafted and wrote the paper with participation of all coauthors. CP contributed in the software development, AN in the development of the database and SAG in the statistical study. IAD and MG were consulted regarding the interpretation of the results. ND, AB and SB contributed to the radiation belt simulations with the Salammbô model and were also consulted regarding the interpretation of the results.

*Competing interests.* The authors declare that they have no conflict of interest.

*Acknowledgements.* This work has received funding from the European Union's Horizon 2020 research and innovation programme "SafeSpace" under grant agreement No 870437. The authors acknowledge the THEMIS/FGM and EFI teams for the use of the corresponding data

sets which can be found online in http://themis.ssl.berkeley.edu/data_products/index.php and the developers of the International Radiation

Belt Environment Modeling (IRBEM) library that was used to calculate the L\* and MLT values via the Olson–Pfitzer quiet model.

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
