# Peer review of "The "SafeSpace" Database of ULF Power Spectral Density and Radial Diffusion Coefficients: Dependencies and application to simulations"

_Annales Geophysicae, 2021_

## Author Comment (AC1)

**Response to reviewers**

**Reviewer #1 Evaluations**:
**Summary**: The work deals with improving the quantification of one of the main processes acting in the radiation belts: radial diffusion. It provides a variety of content of potential importance: It briefly describes how electric and magnetic field measurements from the three THEMIS inner probes (A, D, E) were processed to compute products equated with radial diffusion coefficients, DLLs. It discusses several dependencies of the database related to spatial location and magnetic activity. It compares and contrasts the database outputs with various published models. It also shows two numerical simulations of outer radiation belt dynamics: one where radial diffusion is parameterized by the data products introduced in this manuscript, and the other where radial diffusion is parameterized by the published model that best compares with the database (l.261-262). One of the main findings is that "all models underestimate the DLL during quiet times and at low L* values, while they overestimate the DLL during high levels of geomagnetic activity and at high L* values" (l.279-281).

**General Comments**: The work claims to provide a database of "accurately calculated" radial diffusion coefficients (l.4, l.12, l.71, l.216, l.246, l.265). Yet, it fails to be convincing. A much more rigorous treatment of both data processing and scientific presentation is required to demonstrate the validity and significance of the work.
Response: We agree with the reviewer that the expression "accurately calculated" may lead to misinterpretations. The term "accurate" intended to describe the detailed process we followed for the calculation of the DLL database, from the pre-processing of the data to final scientific product. Nevertheless, we acknowledge that any calculation of the DLL is an estimation based on several assumptions and, thus we have removed this term from the manuscript. Furthermore, we have included in the manuscript a detailed description concerning the entire data processing chain. To that end we have also discussed several assumptions DLL database which arise from the theoretical approach used in this study and the inherent limitations of the in-situ data.

**Specific Comments:**
**Major comments:**
1. The database does not provide radial diffusion coefficients:
A radial diffusion coefficient quantifies the long-term phase-averaged effect of small electromagnetic fluctuations on trapped particles' third adiabatic invariant (e.g., Schulz and Lanzerotti, 1974). Thus, a radial diffusion coefficient is independent of magnetic local time by definition. In this work, the products resulting from THEMIS data processing present significant variations with magnetic local time (section 3.2, Figure 4). This feature is enough to demonstrate that the database does not provide a time series of radial diffusion coefficients.
Response: We would like to thank the Reviewer for noting these points in the calculation of the radial diffusion coefficients. Indeed, the radial diffusion coefficient, DLL, quantifies the mean square displacement of radiation belt electrons across Roederer's L* as a result of fluctuations in the magnetic and electric fields. In the classic electromagnetic diffusion formulas proposed by Falthammar (1965), particle perturbations leading to diffusion result from variations in the magnetic field along the drift orbit and the electric fields induced by these magnetic field fluctuations as well as electric potential fluctuations, DLL,m and DLL,e.

In this manuscript, for the calculation of DLL, we have adopted the newer formulas for radial diffusion coefficients proposed by Elkington et al. (2003) and further developed by Fei et al. (2006) that consist of a component that quantifies radial diffusion driven by magnetic field disturbances in the direction of the background magnetic field, DLLB and a second component that quantifies radial diffusion driven by azimuthal electric field disturbances, DLLE. Since no coupling between wave magnetic and electric fields through Faraday's law is assumed, there are uncertainties introduced in the derivation of radial diffusion coefficients by Fei et al.(2006). We have noted that Lejosne (2019) has estimated that, in the presence of magnetic field disturbances, adopting the approach of Fei et al. (2006) leads to underestimation of the total radial diffusion coefficients by a factor of 2. However, as Sandhu et al. (2020) have suggested and as we demonstrate in section 4.1, this discrepancy is comparatively minor relative to the large variability of the calculated values which span orders of magnitude especially during magnetic storms.

Furthermore, spatial variations in the power of magnetic and electric field perturbations have been found to impart local time dependencies to calculated diffusion coefficients. In the following figure we demonstrate that wave power calculated based on measurements from three spacecraft of the THEMIS constellation is highly dependend on the limited MLT sector sampled.

[Figure]

**Figure 1: Logarithms of the mean ULF power with 1-min as a function of MLT (dMLT=1 hour) and L\* (dL\*=0.1) for three levels of geomagnetic activity: (left column panels) Kp<3, (middle column panels) 3<Kp<5 and (right column panels) Kp>5. Top and bottom row panels correspond to the power in the azimuthal electric field component and in the compressional magnetic field component, respectively.**

Local time variations in wave power indicate sources of wave activity both internal (coupling with ring current ions and substorm particle injections) as well as external (solar wind driving). However, using measurements from a single spacecraft or from a single mission that sample a specific MLT sector can result to under- or over-estimates of radial diffusion coefficients, since spatial variations are neglected. In our case, the maximum MLT coverage from all three spacecraft does not exceed 6 hours per hour and per L\*. This means that our DLL (and of course any other estimated by in-situ measurements) employs a small fraction of the full MLT coverage which would be required. Therefore, figure 4 in the manuscript reflects exactly the features presented in the above mentioned figure 1.

We emphasize that radial diffusion is a drift-averaged process and radial diffusion coefficients should describe an average over all local times and the possibility of combining

measurements from missions and spacecraft sampling different parts of the magnetosphere needs to be explored. Our efforts have been currently focused on quantifying the magnitude of radial diffusion due to ULF waves observed solely by the THEMIS spacecraft since combining measurements from different missions will need intercalibration of measurements which is beyond the scope of this study.

A brief description has been added in the revised section 2 of the manuscript as follows: "Equations 2 and 3 also implicitly assume a uniform distribution of wave power in azimuth. In reality, the azimuthal distribution of the wave power in the Pc4-5 range depends on their generation mechanism, e.g. the wave power due to the Kelvin-Helmholtz instability is expected to be greater near dawn and dusk sectors, while due to the pressure pulses from the solar wind is expected to be greater near noon. Furthermore, the maximum MLT coverage from all three spacecraft does not exceed 6 hours per hour and per L*. This means that our DLL--and of course any other estimated by in-situ measurements [Jaynes et al. 2018, Olifer et al. 2019, Sandhu et al. 2021]--employs a small fraction of the full MLT coverage which would be required. We note that radial diffusion is a drift-averaged process and radial diffusion coefficients should describe an average over all local times. Nevertheless, in order to achieve a full MLT coverage, one would need a large multi-satellite dataset which would span several years. Our efforts have been currently focused on quantifying the magnitude of radial diffusion due to ULF waves observed solely by the THEMIS spacecraft since combining measurements from different missions will need inter-calibration which is beyond the scope of this study."

Another paragraph has been added in section 3.2 describing the physical meaning of figure 4 as follows: "We must note that the aforementioned MLT dependence reflects directly the azimuthal distribution of power for both the magnetic and the electric component of the DLL. This means that even though the radial diffusion coefficient is calculated with the drift-averaging assumption, in practice, the limited MLT coverage from single mission in-situ data introduce an azimuthal structure, which accounts for the coupling of external and internal ULF generation mechanisms and may be quite important for future modelling efforts."

Moreover, we have added a paragraph in the end of section 4.1 discussing how the aforementioned assumptions could affect the comparison of our database with the results of the semi-empirical models.

Finally, we have clarified in section 2 that our database consists of two parts: a) the ULF wave PSD, which is stored in daily CDF files with 1-min resolution for each spacecraft separately, and b) the drift-averaged DLL (grouped in bins with dt=1 hour and dL*=0.1).

2. THEMIS data processing, and its presentation, need improvement:
- Quantifying radial diffusion using satellite measurements is a challenging task. For instance, it requires differentiating spatial and temporal variations from a time series of field measurements sampled along spacecraft trajectory, often in the presence of strong spatial gradients. How this is achieved remains unclear.
Response: As mentioned in the previous comment response, we have included in the manuscript a detailed description concerning the entire THEMIS data processing chain.

Concerning the spatial gradients, we are not entirely sure what exactly the reviewer is referring to. We have included a paragraph in section 2.1 which refers to the magnetic field gradients as the spacecraft moves close to the Earth. This paragraph is as follows: "Note that as the satellites move inbound and outbound with high velocities at low L-shells, the

magnetic field measurements exhibit, not only orders of magnitude increase, but very large gradients as well. These large gradients make it quite difficult to estimate the background trend, which has to be removed. Even if we filter the magnetic field time-series, the filtered signal's amplitude still grows significantly near perigee, which renders any PSD calculations erroneous. Therefore, we manually remove the corresponding part of the spectrum."

Furthermore, Fei's expressions are based on the assumption that the asymmetric background magnetic field leads to enhanced radial diffusion in the presence of broadband ULF waves. Following previous studies (Ozeke et al., 2012, Ali et al., 2015, Liu et al., 2016, Jaynes et al.,2018 and others), we have determined the power spectrum of ULF waves along each spacecraft orbit but, due to the limited coverage of THEMIS spacecraft measurements, it is not possible to determine the waves mode structure. In this light, we opted to use the weighted averaged power over the whole frequency range under study (i.e. Pc4-5 frequency range) in the place of waver power at a specific frequency. This procedure and its benefits are discussed in detail in section 2.2.

Finally, in order to extract field perturbations in the Pc4-5 frequency range, the background electric and magnetic field was identified by taking a running average over a 30 min sliding window (see also section 2.1). The power of magnetic field perturbations was calculated in the direction of the background magnetic field (compressional perturbations) and power in local electric field perturbations in the azimuthal direction. However, due to spacecraft motion, separation between spatial and temporal variations is not possible. Temporal variations may be introduced by the dynamics of the different wave sources and these have not been separated from spatial variations of ULF wave components due to weakening of the local magnetic field strength associated with an enhanced ring current population or an increase in the local plasma mass density.

- Fei et al. formulas apply at the magnetic equator only.  Yet, THEMIS probes do not necessarily sample the magnetic equator. The manuscript does not explain how this feature is taken into account in the data processing.
Response: As mentioned in the revised section 2, Fei's approach considers equatorially mirroring particles only, while THEMIS satellites do not necessarily sample the magnetic equator. Nevertheless, they remain very close to the magnetic equator throughout their trajectories in the outer belt [Angelopoulos 2008, Turner et al. 2012] something that allows us to assume that the uncertainty in the DLL calculation will be rather small. This is also supported by the results shown in the following figure where we have plotted the mean power versus the Beq/Blocal ratio for different values of L*. As shown, both in the electric and magnetic field power, the variation of power versus Bratio is up to a factor of 2, at least for Bratio values larger than 0.8 (note that this threshold in Bratio is used in our study so all results correspond to DLL values at points with Bratio>0.8). Nevertheless, we should note that there is no straightforward comparison with the dataset used by Sandhu et al. 2020, since we have no information about whether they have sorted their dataset based on magnetic latitude or about the model used for the calculation of the magnetic ephemeris data.

Note that we have revised the corresponding paragraph of the manuscript as follows:
"Finally, we emphasize the fact that our results on the MLT asymmetry are in good agreement with Sandhu et al. 2020 who used Van Allen probes data (different magnetic latitude) to infer the radial diffusion coefficients. This agreement also indicates that the uncertainty introduced by the magnetic latitude (and already discussed in section 2) is insignificant, even though there is no straightforward comparison with the dataset used by

latter authors, since we have no information about whether they have sorted their dataset based on magnetic latitude or about the model used for the calculation of the magnetic ephemeris data."

[Figure]

Figure 2: Logarithms of the mean ULF power with 1-min as a function of Beq/Blocal (dB=0.05) for 5 L* bins with dL*=0.1.

- Not all choices made during data processing are well explained or well justified. For instance, why the equation (4)? Why "cdelta" =0.76? What is the definition of "dj"?
Response: Indeed the data processing section was insufficient. Please note that we have revised section 2 in order to provide many more details to the reader concerning the entire data processing chain. Especially for the calculation of the ULF wave PSD, we have used the wavelet analysis as described in Torrence and Compo [1998]. The latter authors provide an in-depth analysis of the use of wavelet functions. This analysis is quite long and we believe that including it in the manuscript would be out of the scope of our study and would render the reading of the manuscript unnecessary difficult. Nevertheless, we have included the definitions of specific parameters.

For example, dj is the sampling scale of the wavelet analysis, which of course depends on the frequency range under study. Cdelta is a smoothing factor which depends on the non-dimensional frequency $\omega_0$ of the Morlet wavelet:

$$\psi_0(\eta) = \pi^{-\frac{1}{4}} \cdot e^{i\omega_0\eta} \cdot e^{-\frac{\eta^2}{2}}$$

where η is a non-dimensional time parameter. For the Morlet wavelet $\omega_0$ is taken as 6 to satisfy the admissibility condition (Farge 1992) and then Cdelta is empirically derived as 0.776.

- Farge, M., 1992: Wavelet transforms and their applications to turbulence. Annu. Rev. Fluid Mech., 24, 395–457.
- Torrence, C., and Compo, G. P.: A practical guide to wavelet analysis. Bulletin of the American Meteorological Society, 79(1), 61–78, 1998.

3. The claim that the data products are accurate is not justified:
Before being able to make any claim regarding the accuracy of the approach, it seems necessary to discuss the extent to which the outputs depend on the variety of choices made during data processing. Yet, this has not been done.
Response: As stated before the term "accurate" intended to describe the detailed process we followed for the calculation of the DLL database, from the pre-processing of the data to final scientific product, and in order to avoid any misinterpretations we have replaced it in the manuscript. Also we have included the subsection 2.3 (Assumptions) where we discuss

all the assumptions used for the calculation of the PSD and DLL including the theoretical approach (Fei et al 2006) and the inherent limitations of our dataset. Furthermore, all the results presented in this study have been further discussed in the basis of these assumptions.

**Minor comments:**

* Table 1: Units are missing. The list of limitations provided is incomplete.

Response: Duly amended. We have now revised the table in order to included L* and Kp limitations for each model.

* While Falthammar's (1965) framework was developed in the non-relativistic case, the extension to relativistic particles is straightforward (e.g. Schulz and Eviatar, 1969). Thus, the claim that Falthammar's formulation is "valid for sub-relativistic particles, only" (l.32) is misleading.

Response: We thank the reviewer for the valuable information. The sentence has been removed from the manuscript.

**References:**

Schulz and Eviatar (1969), Diffusion of equatorial particles in the outer radiation zone, https://doi.org/10.1029/JA074i009p02182

Schulz and Lanzerotti, 1974, Particle Diffusion in the Radiation Belts, https://doi.org/10.1007/978-3-642-65675-0

---

## Author Comment (AC2)

**Response to reviewers**

**Reviewer #2 Evaluations**:
**Summary**: This work sets out an analysis of the SafeSpace electron diffusion coefficients database, comparing the magnitude of the magnetic and electric parts (relating to the formulation of Fei et al., 2006) under different environmental conditions parameterised by a number of indices. The method of calculating DLL using data from the THEMIS spacecraft is first described, followed by the main analysis. The time evolution of the ratio between DLLB and DLLE is discussed, and a comparison with various empirical models of DLL is then made, followed by an application of the SafeSpace coefficients using a physical model in order to simulate the outer electron belt over a month-long period. As part of the conclusion, the manuscript makes the claim that empirical models tend to underestimate DLL "at low levels of geomagnetic activity at all L*" based on results of the earlier comparison, and that DLLB can reach values comparable to, or in excess of, DLLE for periods following ICMEs.

**General Comments**: The authors show some interesting details about the time evolution of DLL following geomagnetic disturbances (Fig. 5). The authors also show correlations between changes in DLL and various indicies, and relate this to the physical processes driving each index (Fig. 3, 4). However, the method for calculating DLL from spacecraft data for the SafeSpace database is not clearly explained, and the authors do not clearly present evidence to support later claims that empirical models under/over-estimate just because they do not agree with SafeSpace. Some further discussion about the MLT-dependence of provided DLL is also required.

**Major comments:**
1. Section 2
As the authors are aware, DLL is used to quantify the time evolution of phase space density over many drift orbits due to small, repeated electromagnetic fluctuations. In this work DLL is presented as MLT-dependent (or perhaps, the authors' method of calculating DLL from THEMIS data is sensitive to the MLT at which data was collected). In any case, Section 2 should address the physical meaning of the MLT dependence of SafeSpace DLL, and where this arises from.
In addition, Section 2 jumps directly into describing measurements, without describing what these measurements are being used for. In general, the structure of Section 2 should be reworked. As a starting point, an example way to order things might be:
DLL was calculated directly from measurements in order to construct a database parameterised by solar wind and geomagnetic parameters. To calculate DLL, we used an approach based on the Fei et al. (2006) formulation. This approach involved considering "the compressional component of the magnetic field…" . The calculation of DLLB and DLLE depends on … These parameters were first determined using measurements from the THEMIS satellite.
[Then, describing how:] "We use 4-sec resolution measurements of the magnetic field vector…" "Complementary 1-min measurements of solar wind…" (etc.) "The THEMIS magnetic and electric field data were pre-processed by transforming them into a Mean 90 Field Aligned (MFA) coordinate system…" etc. [Then elaborate on the data processing method in more detail to put the MLT dependence in context.]
Response: We thank the reviewer for the suggestions. Indeed the data processing section was insufficient. Please note that we have revised section 2 according to the aforementioned suggestions in order to provide many more details to the reader concerning

the entire data processing chain. To that end we have also discussed several assumptions of our DLL database which arise from the theoretical approach used in this study and the inherent limitations of the in-situ data.

Especially, concerning the MLT dependence of the DLL, we have discussed that it reflects directly the azimuthal distribution of power for both the magnetic and the electric component. As we now state in section 2: "Equations 2 and 3 also implicitly assume a uniform distribution of wave power in azimuth. In reality, the azimuthal distribution of the wave power in the Pc4-5 range depends on their generation mechanism, e.g. the wave power due to the Kelvin-Helmholtz instability is expected to be greater near dawn and dusk sectors, while due to the pressure pulses from the solar wind is expected to be greater near noon. Furthermore, the maximum MLT coverage from all three spacecraft does not exceed 6 hours per hour and per L*. This means that our DLL--and of course any other estimated by in-situ measurements [Jaynes et al. 2018, Olifer et al. 2019, Sandhu et al. 2021]--employs a small fraction of the full MLT coverage which would be required. We note that radial diffusion is a drift-averaged process and radial diffusion coefficients should describe an average over all local times. Nevertheless, in order to achieve a full MLT coverage, one would need a large multi-satellite dataset which would span several years. Our efforts have been currently focused on quantifying the magnitude of radial diffusion due to ULF waves observed solely by the THEMIS spacecraft since combining measurements from different missions will need inter-calibration which is beyond the scope of this study."

The aforementioned paragraph implies that any attempt to estimate the DLL from in-situ data (independent of the mission used) will introduce an MLT dependence due to the limited azimuthal coverage, which however may be important for future modeling efforts.

2. Section 4
The authors discuss the underestimation/overestimation by empirical models compared with the SafeSpace DLL. However, I feel it is important for the authors to also discuss the uncertainty in the SafeSpace DLL that may also be a cause of disagreement versus empirical models. This would strengthen the author's claim that the SafeSpace DLL are accurate.
Response: We thank the reviewer for the constructive comment. We have now included a paragraph at the end of section 4.1 discussing these uncertainties and how they could affect our results. The paragraph is as follows:

"Finally, we have to consider the uncertainties in the SafeSpace calculated DLL that may also be a cause of disagreement versus the aforementioned semi-empirical models. As discussed in section 2, it has been shown that the Fei et al. 2006 approach can underestimate the radial diffusion coefficient by a factor of two compared with the Falthammar 1965 approach. This is sufficient to explain the difference exhibited by our DLL and the Brautigam & Albert 2000 and Boscher et al. 2018 models at L*>4, but it cannot explain the up to a factor of 10 difference at lower L-shells. Another uncertainty, also discussed in section 2, comes from the limited MLT coverage of THEMIS satellites used in this study. Nevertheless, the results of figure 6 are averaged values of the SafeSpace DLL for specific values of L* and Kp over 9 years of calculation, thus including several MLT values."

We have further added significant discussion in section 4.1 discussing possible sources of the disagreement between the SafeSpace DLL and the various semi-empirical models.

**Minor comments:**

1. first paragraph

The first paragraph is vague. It is important to 'draw the reader in' at this stage. The authors could cut the first sentence down to something like, "The outer radiation belt exhibits electrons at energies from a few hundred keV to several MeV [reference]."

Then the authors could go straight to the topic of radial diffusion, e.g.: "Radial diffusion has been established as one of the most important mechanisms causing energization [references] and loss [references] of relativistic electrons."

Response: We have now changed this paragraph as follows: "The outer radiation belt exhibits electrons at energies from a few hundred keV to several MeV [Daglis et al. 2019]. Radial diffusion has been established as one of the most important mechanisms that contributes to this broad energy range of electrons since it can lead to both energization [Jaynes et al. 2015, Li et al. 2016, Katsavrias et al. 2019a, Nasi et al. 2020] and loss of relativistic electrons [Morley et al. 2010, Turner et al. 2012, Katsavrias et al. 2015, Katsavrias et al. 2019b]."

2. second paragraph

Again, the first sentence could be omitted. The second sentence could be expanded on like so:

"Ultra-Low Frequency (ULF) waves in the Pc4-5 band (1 and 22 mHz) can violate the third adiabatic invariant L* of..." Next sentence: "This drives radial diffusion by..."

This way the explanation comes first.

Response: We have now changed this paragraph as follows: "Ultra-Low Frequency (ULF) waves in the Pc4-5 band (1--22 mHz) can violate the third adiabatic invariant L* of the energetic electrons. This drives radial diffusion by conserving the first two adiabatic invariants under the drift resonance condition $\omega = m\omega_d$, where $\omega$ is the wave frequency, m is the azimuthal wave mode number and $\omega_d$ is the electron drift frequency [Elkington et al. 2003]."

3. line 34

The authors might consider replacing "something that runs counter to basic physical concepts of electromagnetism" with a more specific summary of the limitation. It is described well by Lejosne, 2019, and this paper is referenced in line 113. It can be referenced here as well.

Response: A brief explanation based on Lejosne (2019) has been included in the corresponding paragraph.

4. line 51

Replace "limitations of" with "dependence on" (and in line 219)

Response: Duly amended.

5. a general comment about repetition

The word "moreover" beginning line 58 is repeated at the beginning of the next sentence, and it is also in the previous paragraph. Use an alternative here, for example, "Furthermore, observed DLL have been shown...". It can be omitted in the next sentence too, for example: "Several case studies have demonstrated..."

"Moreover" is used yet again on line 66, try instead something like:

"...overestimated by the empirical model of Ozeke et al. (2014). At times, the difference between empirically modelled values and event-specific diffusion coefficients was shown to be multiple orders of magnitude."

The word "Nevertheless" is also used three times from line 100 to 115. As before, it would read better with a substitute.

Response: All sentences have been modified in order to avoid repetition.

6. line 75
There is a section 5 too, so don't use the word "finally" to describe section 4.
Response: Duly amended.

7. Figure 1
The Figure 1 font is difficult to read, try something like Arial, Calibri, etc. and use a darker red background for the white text.
Response: The figure colors and fonts have been modified.

8. line 139
The authors show that the CC between Psw and DLLE is weak. They then state that changes in Psw "are not really linked with the electric DLL component", yet mention that they are an important ULF wave generation mechanism. Are they implying the generation of ULF waves is not related to changes in DLLE? Be specific about this, rather than saying "not really linked".
Response: We apologize for the poor choice of words in this sentence. We have revised the manuscript as follows: "A possible explanation of this feature could be that, since solar wind pressure pulses produce mainly global magnetospheric oscillations [Kepko et al. 2002,Takahashi et al. 2012], they do not affect the azimuthal electric field variations and thus the electric DLL component."

9. line 152
The distribution is shown in terms of the magnetic coordinate L*, rather than a spatial coordinate. So if this is a spatial distribution, does it relate to the magnetic equator, or is there just no dependence on magnetic latitude, etc? Line 178 implies the dependence on latitude is weak, but, this should be clarified.
Response: As mentioned in the revised section 2, Fei's approach considers equatorially mirroring particles only, while THEMIS satellites do not necessarily sample the magnetic equator. Nevertheless, they remain very close to the magnetic equator throughout their trajectories in the outer belt [Angelopoulos 2008, Turner et al. 2012] something that allows us to assume that the uncertainty in the DLL calculation will be rather small. This is also supported by the results shown in the following figure where we have plotted the mean power versus the Beq/Blocal ratio for different values of L*. As shown, both in the electric and magnetic field power, the variation of power versus Bratio is up to a factor of 2, at least for Bratio values larger than 0.8 (note that this threshold in Bratio is used in our study so all results correspond to DLL values at points with Bratio>0.8). Nevertheless, we should note that there is no straightforward comparison with the dataset used by Sandhu et al. 2020, since we have no information about whether they have sorted their dataset based on magnetic latitude or about the model used for the calculation of the magnetic ephemeris data.

Note that we have revised the corresponding paragraph of the manuscript as follows: "Finally, we emphasize the fact that our results on the MLT asymmetry are in good agreement with Sandhu et al. 2020 who used Van Allen probes data (different magnetic latitude) to infer the radial diffusion coefficients. This agreement also indicates that the uncertainty introduced by the magnetic latitude (and already discussed in section 2) is insignificant, even though there is no straightforward comparison with the dataset used by latter authors, since we have no information about whether they have sorted their dataset

based on magnetic latitude or about the model used for the calculation of the magnetic ephemeris data."

[Figure]

Figure 1: Logarithms of the mean ULF power with 1-min as a function of Beq/Blocal (dB=0.05) for 5 L* bins with dL*=0.1.

10. line 170
"On the other hand, the observed asymmetry in the electric component indicates that DELL is not only linked with solar wind speed but with internal mechanisms such as substorm activity, something that is also in agreement with the results of figure 2. "
Explain briefly why the asymmetry indicates a link with internal mechanisms.
Response: The intention of this sentence was to highlight the high values of DLLE at the nightside sector (0<MLT<3) shown in the upper panels of 4.

We have revised the corresponding text as follows: "On the other hand, the observed asymmetry in the electric component indicates that DLLE is not only linked with solar wind speed but with internal mechanisms such as substorm activity, especially during quiet or moderate magnetospheric activity. This is supported by the remarkable agreement of the DLLE MLT distribution (top row panels of figure 4) with Nose et al. 1998, who stated that substorms generate azimuthal ULF fluctuations at the nightside which peak at 1--2 MLT. Furthermore, this is also in agreement with the results of figure 2 and the significant correlation of DLLE with the AE index."

11. line 157 and line 190
"exceeds the value of 10…"
"a median value of 1000…"
Remember to always state units throughout
Response: Duly amended.

12. line 204 - 205
"up to two orders of magnitude compared with the magnetic component."
State that it is the ratio between the two which varies, from X up to ~100, etc.
Response: Duly amended.

13. line 212
"Also note that this feature present during SIR disturbances as well."
Where on Figure 5 is this shown? It's not as obvious as the change during ICMEs.
Response: We have revised this sentence as follows:

"Note that this feature, even though it is no that obvious, may be important during SIR disturbances as well. As shown in the bottom right panel of figure 5, the DLL ratio at $L^*>5.5$ is decreased from approximately 100 to approximately 1 at +-3 hours from t0. We suggest that this difference in the DLL ratio between ICME and SIR--driven disturbances is probably attributed to the existence (or not) of shocks, which produce significant increase of the dynamic pressure and accompany, more often, the ICME--driven events."

14. end of Section 3

In Section 3.1, line 124, the authors explain how the energy/first invariant dependence of DLL does not significantly change the CCs shown in Figure 2. Can the same be said for the ratio of DLLE to DLLB shown in Figure 4, for example, if DLLB is energy dependent, does it still increase above DLLE at other energies? A few words addressing this would be sufficient. However it is also necessary to elaborate on the following:

"Furthermore, at $L^*> 6$, the DBLL is comparable to the DELL up to approximately 12 hours after t0."

I am having trouble seeing this on Figure 5. What is meant by "comparable?" The conclusion on line 214 that DLL becomes energy dependent due to higher DLLB is only valid during the period following a disturbance, I presume. Therefore, it should be made clear how long this lasts, and what is the ratio of DLLE/DLLE, in order to show the reader this effect is important for radiation belt simulations.

Response: The following sentence has been added concerning the energy dependence of the DLLB:

"Furthermore, this feature is expected to be dependent on the first adiabatic invariant as well, since greater values of μ produce greater values of DLLB, which will consequently lead to changes in DLL ratio. It is also expected that, except the magnitude, the change in μ will affect both the duration and the $L^*$ coverage of this feature as well. In a future study we intend to investigate in greater detail these changes."

We have further clarified the duration of the DLLratio<1 as follows:

"This feature changes dramatically during ICME driven disturbances and around +- 6 hours from the maximum compression of the magnetetopause where the DLL ratio decreases below 1 at all $L^*$ values. Furthermore, at $L^*>6$, the DLL ratio is approximately 1 up to 12 hours after t0."

15. section 4 / figure 7

"As shown in the 500 keV electron energy, simulation results exhibit more injections at high $L^*$ ($4<L^*<5.5$) both during the relatively quiet period on early March and during the intense St. Patricks storm when using the calculated DLL…"

The injection events in Figure 7 seem to correspond to an external source of particles becoming trapped due to magnetic variability. Does this process involve diffusion? I am not sure why the different DLL leads to more injections.

The SafeSpace results do seem to show some improvement, but it would be better if the authors also addressed the disagreement between the MagEIS data and Salammbo results in either case, since it appears to be significant. I assume this disagreement is not just due to DLL, but rather a number of modelling factors.

Response: We thank the reviewer for the constructive comment. Indeed the word "injection" may have been misleading since it is more often than not used to describe the substorm injections. Here we are referring to transport/diffusion of low energy electrons from higher L-shells (beyond GEO), which of course is often combined with substorm

injections. Therefore, we have replaced the term "more injections" with "more intense radial transport".

The disagreement between Salammbo and MagEIS data comes rather from the fact that we have not used the energy diffusion term in the simulations. As reported in several studies, the in-situ acceleration due to energy diffusion via chorus waves is usually more important for the enhancement of 1-2 MeV electrons. Therefore we have added the following sentence in the manuscript: "We note that the magnitude of the flux in the Salammbo simulation is not expected to agree with the MagEIS data due to the lack of the energy diffusion term (in-situ acceleration by VLF chorus waves), which for the St. Patrick's event of 2015 has been shown to be crucial especially for 1-2 MeV electrons (Li et al. 2016).

- W. Li, Q. Ma, R. M. Thorne, J. Bortnik, X.-J. Zhang, J. Li, D. N. Baker, G. D. Reeves, H. E. Spence, C. A. Kletzing, W. S. Kurth, G. B. Hospodarsky, J. B. Blake, J. F. Fennell, S. G. Kanekal, V. Angelopoulos, J. C. Green, and J. Goldstein. Radiation belt electron acceleration during the 17 March 2015 geomagnetic storm: Observations and simulations. Journal of Geophysical Research (Space Physics), 121:5520–5536, June 2016. doi: 10.1002/2016JA022400.

---

## Referee Report (RR1)

General comments:

The authors have implemented all the revisions requested, and significantly improved the structure and clarity of the discussions. The results are interesting and important, and I recommend this paper for prompt publication.

If possible, I recommend the following grammatical / typographical revisions, but I think these can be implemented in the copyediting stage and should not hold up publication.

Recommended grammatical / typographical revisions:

Line 95: "quite similar with"
Change to "quite similar to"

Line 103: "field measurements exhibit, not only…"
Delete comma, change to: "field measurements exhibit not only…"

Line 107: "Especially for the magnetic component…"
Delete "Especially", change to" "For the magnetic component"…

Line 348: "the relativistic electron population and but generally…"
Delete "and", change to: "the relativistic electron population, but generally…"

Line 361: "dependence of these calculated DLL to several solar wind…"
Replace "to" with "on", change to: "dependence of these calculated DLL on several solar wind…"

---

## Author Response (AR2)

**Response to reviewers**

**Reviewer #1 Evaluations**:
I thank the authors for the reply. However, my most major comments remain valid:

1. The database still does not provide radial diffusion coefficients. A radial diffusion coefficient is independent of magnetic local time, yet the database provides quantities that clearly depend on MLT (Fig.4).
The answer to this comment seems to be that "in order to achieve a full MLT coverage, one would need a large multi-satellite dataset which would span several years" (l.150-151), and since this study relies "solely" (l.152) on THEMIS data, the MLT coverage is limited.
=> This explanation is confusing. This study relies on 9 years of multi-point magnetic and electric field measurements by THEMIS A, D and E (l.3-4). Thus, it provides full MLT coverage. Drift-averaged radial diffusion coefficients can be obtained from this data by averaging PSDs (phase space density) over all MLT (magnetic local time) bins. Yet, this has not been done.

In Sandhu et al.'s study (2021), how the PSD varies with MLT is discussed, but the radial diffusion is then calculated by averaging over all MLT bins and performing superposed epoch analysis. In other words, Sandhu et al., who computed DLL using Van Allen Probes data only, do not claim to provide "event specific" DLL.

It seems that the objective of this study is to provide "event-specific" radial diffusion coefficients, using solely THEMIS data, assuming "uniform distribution of wave power in azimuth" (l.143). This is problematic. The data analysis clearly shows that this assumption is not valid (wave power depends on MLT): This demonstrates that the data products are, at best, poor estimates of the magnitude of radial diffusion. Under these circumstances, the objective of providing "event-specific" radial diffusion coefficients using solely in-situ spacecraft measurements appears unrealistic.
To address this comment, I would suggest to either:
- Remove the time series of data products equated with radial diffusion coefficients in the database, and compute statistical DLLs, following an approach similar to what Sandhu et al.'s did

- Or Rename the data products using a totally different terminology, so that they are not mistaken with radial diffusion coefficients by a hasty reader.

In this case, it would still be interesting to see the DLLs that would result from averaging over all MLT bins.
In all cases, Figure 4 should be removed, together with any discussion of DLL with MLT. In addition, the manuscript should not claim to provide "event specific" DLLs.

Our response: As we have already thoroughly discussed in the revised manuscript: "Even though we have followed a well-established methodology in order to calculate - as accurately as possible - the ULF PSD and the corresponding DLL there are still worth-mentioning assumptions, which are based on the theoretical approach we have used as well as on the inherent limitations of the in-situ data". Of course, this is true not only for the DLL calculations but for every (without exceptions) work attempting to combine theoretical approaches with actual observations in the field of space physics. In this case, an important

assumption is that we infer DLL using a small fraction of MLT coverage provided by the three THEMIS spacecraft.

The reviewer refers to Sandhu et al, 2021 indicating "the radial diffusion is then calculated by averaging over all MLT bins and performing superposed epoch analysis". We believe that there is a misunderstanding here. It is quite obvious that the methodology followed by Sandhu et al. is similar to ours. Sandhu et al. also discuss time-series of DLL (see figure 1 in Sandhu et al. 2021). Of course, it is impossible for these DLL to be drift averaged, simply because of the limited MLT coverage of RBSP. Then, indeed, a superposed epoch analysis is performed but only in order to highlight statistical features. There, the authors **assume** that, since several events are considered, there is a full MLT coverage by RBSP satellites. We have adopted exactly the same assumption in figure 5 of our manuscript, where we compare ICME and SIR driven events using superposed epoch to reveal statistical features. Note that even figure 5 in Sandhu et al., which shows the median of the superposed epoch analysis, also shows DLL time-series in the background. Regarding DLL time-series, calculations similar to our own have also been published in several recent papers. Examples include Olifer et al. 2019 (see figure 6 in the corresponding paper) and Jaynes et al. 2018 (see figure 4 in the corresponding paper). The aforementioned examples, and our own work, highlight the fact that, even though the MLT coverage of in-situ data cannot provide the ideal full azimuthal distribution, this is the only way to calculate in-situ time-series of DLL.

We would like to stress that, unlike several other works, we have already included: a) a dedicated subsection stressing out all the assumptions of our approach and b) extensive discussion on the limitations of this approach. Therefore, we find the suggestion of the reviewer to rename the DLL product "so that they are not mistaken with radial diffusion coefficients by a hasty reader", unjustified. However, in the interest of accuracy, we have replaced the sentence in line 113 "These drift-averaged radial diffusion coefficients" with "These grouped radial diffusion coefficients". For the same reason, we have replaced the term "event-specific DLL" with "DLL time-series", throughout the entire manuscript.

The reviewer alternatively suggests to: "Remove the time series of data products equated with radial diffusion coefficients in the database, and compute statistical DLLs, following an approach similar to what Sandhu et al did". Sandhu et al. derive statistical DLLs which are calculated as median values over 45 storms and these DLLs span approximately 2 orders of magnitude, while the actual DLL time-series can span 6 orders of magnitude (see also figure 5 panel d in Sandhu et al). This plainly demonstrates the fact that a statistically inferred DLL using the superposed epoch analysis will overestimate/underestimate the actual DLL time-series by several orders of magnitude, simply due to the long-period averaging, and thus, will be a less favorable option for use in modeling/simulations. Such a feature is even more pronounced in statistical studies that use median values of entire datasets over a long time period (e.g. Ali et al 2016, etc) and an important disadvantage of most semi-empirical models. Indeed our work, and others (e.g. Olifer et al. 2019; Sandhu et al. 2021; Drozdov et al. 2020), have shown that these empirical models can underestimate the DLL by orders of magnitude especially during intense geomagnetic activity. All the above indicate that statistically inferred DLLs with the use of superposed epoch analysis would be – as the reviewer states – unrealistic for use in simulation/modelling efforts.

Furthermore, we have explicitly stated in the revised manuscript that: "…the MLT dependence reflects directly the azimuthal distribution of wave power for both the magnetic and the electric component of the DLL. This means that even though the radial diffusion coefficient is calculated with the drift-averaging assumption, in practice, the limited MLT

coverage from single mission in-situ data introduce an azimuthal structure, which accounts for the coupling of external and internal ULF generation mechanisms and may be quite important for future modelling efforts". We have also included in the response to the reviewers, the corresponding distribution using ULF wave power instead of DLL only to show that they are quite similar.

[Figure]

Finally, the reviewer states: "This demonstrates that the data products are, at best, poor estimates of the magnitude of radial diffusion". We believe that this is an overstatement and rather degrading of our work. Every attempt to calculate diffusion coefficients is nothing more than an estimation and all the approaches that have been developed and used have their advantages and disadvantages, without any exception (see for example discussions in Brautigam and Albert, 2000; Ozeke et al. 2014 and Lejosne, 2020). We emphasize that we are showing, using actual simulations with the established Salammbo code, that our DLL time-series, not only are comparable with widely used empirical models (i.e. the Boscher model also used in Salammbo simulations) but can also produce much better and meaningful results (see also figure 7 in the revised manuscript). Therefore, we consider this work a worthy contribution to the space physics community, compared with statistically inferred DLL and semi-empirical models.

2. The data processing remains unclear:
Quantifying radial diffusion requires differentiating between: (a) the time variations of the fields that are faster than trapped particles' drift period (responsible for non-adiabatic effect), (b) the slow field variations, occurring on a time scale slower than the drift period (responsible for adiabatic effect) and (c) the spatial variation of the fields (responsible for the trapping). Field variations measured by a spacecraft along its trajectory include all 3 sources of variations. The first component, (a), is the only component of interest for radial diffusion (i.e. we only need to use the time variations of the fields that are below energetic particles drift frequency when computing the PSD), and this component depends on the population energy (e.g. Falthammar, 1968). The manuscript does not clearly explain how this component is isolated, and it does not quantify the error accompanying the estimate.

Our response: Indeed the time variations of the fields that are faster than the trapped particles' drift period are the important component for the calculation of the DLL. As thoroughly discussed in the revised manuscript, those are isolated from the slow field variations by de-trending the time-series using a 20-min moving average which is similar to a low-pass filtering with a cut-off frequency at ~0.8 mHz. This means that we keep oscillations with frequencies higher than 0.8 mHz, which are the ones responsible for the breaking of the

third adiabatic invariant of relativistic electrons. We note that since the spatial variations, especially at the radial distances corresponding to the outer belt, are usually slow variations, they are also filtered out by this procedure. In the following figure we provide an example using the Swarm-A magnetic field data and the Swarm-based CHAOS-7 Magnetic Field Model (Finlay et al. 2020), which can model the internal magnetic field (core and lithosphere contribution) with very high accuracy, thus allowing us to remove the contribution of spatial gradients. The top panel (a) shows the total magnetic field by Swarm-A, and panel b the two processed signals, one after high-pass filtering only and the other after first subtracting the CHAOS model and then filtering. There are some minor differences, but they are almost inconsequential. Panel c shows the PSD of for the high-passed filtered time-series, while panel d, for the high-passed filtered residual after subtraction of the CHAOS model. Finally, bottom panel shows the weighted average wavelet power at all frequencies for both the high-passed only and for the high-passed filtered residual after subtraction of the CHAOS model. As illustrated, even for a low altitude satellite, such as Swarm-A, which is orbiting at approximately 400 km, the frequency content of the signal is almost identical, regardless of the subtraction of the CHAOS model, which is used to remove the spatial variations of the field. For satellites at higher altitudes (such as THEMIS used in our study), where the magnetic field gradients are not nearly as steep, we can safely argue that applying a high-pass filter is enough to remove the variations that are due to the satellite flying through an inhomogeneous magnetic field.

Nevertheless, the spatial variations of the field at very low L-shells can occasionally introduce errors in the estimation the background trend. Thus, wherever needed, we have completely removed the part of the spectrum, where steep spatial gradients exist, manually.

Finally, since we have adopted the Fei et al. 2006 approach we note that the energy dependence is included in the mathematical form of the DLLB component (see also equation 2 in the revised manuscript). We emphasize that, as discussed by Lejosne and Kollman [2020]: "Although Fei et al.'s formalism is inadequate from a theoretical standpoint, it is very convenient from a practical standpoint. It is indeed difficult to differentiate the induced and electrostatic components of an electric field measurement. This poses a serious problem when it comes to applying Fälthammar's formalism to quantify radial diffusion. The same problem is circumvented when applying Fei et al.'s erroneous formalism." To that end we have included both a brief discussion and the corresponding reference.

[Figure]

l.91, regarding the MFA: "The unperturbed field [is] obtained by a 30-min running average of the fields magnitude" + l. 95: " the toroidal and compressional component of the electric and magnetic field, respectively, were detrended using a 20min moving average": is the data averaged twice?

Our response: The compressional component of the field is obtained by the following mathematical form:

$$B_{com} = \Delta B \cdot \frac{B}{|B|} = (B - \bar{B}) \cdot \frac{B}{|B|}$$

where $\bar{B}$ is the unperturbed field obtained by the 30-min running average. The 20-min running average is then used in Bcom and Etor, as clearly described in the revised manuscript, in order to isolate the fast from the slow field variations. Note that the corresponding description has been revised to clarify the transformation (lines 90-96 of the revised manuscript).

l.98: "PSD of the waves in the 2-25mHz" => how is the PSD adapted to the energy of the population (or first adiabatic invariant)?

Our response: According to Fei et al. the energy of the population is included in the magnetic component of the DLL and the mathematical form is shown in equation 2 of the revised manuscript.

l.101: "4s PSDs were averaged in 1 min windows" => what is the motivation behind this step?

Our response: This step accounts for the economy of the dataset. A 4-sec resolution dataset for 9 years would result to ~71,000,000 data points per satellite, while a 1-min resolution dataset has ~4,700,000 data points per satellite. As one can imagine, the latter is less CPU intensive and makes it easier to perform statistical studies. Moreover, the 1-min is chosen because the THEMIS satellites exhibit negligible ΔL and ΔMLT in this timespan.

l.105-109: regarding the removal of spatial field gradients: this seems to be the motivation underlying the fact that PSDs estimates are not provided at low L shells (below L =3). Yet, errors due to the spatial variations of the fields (component (c)) occur also at high L values: The error only increases with decreasing L. What's the level of error created by the spatial variations of the field? And how does it vary with L? This needs to be included in the manuscript so as to demonstrate that L = 3 is the correct cut-off value, and so as to quantify the error in the PSD.

Our response: We would like to highlight, as it has already clarified in the lines 105-108 of the revised manuscript, that we have not chosen L=3 as a cut-off value. Actually, the PSD calculations in the database cover L values down to L~2. In, detail, the part of the spectrum (in radial extend) in each satellite's orbit which, is dominated by the spatial gradients, depends on the level of geomagnetic activity. During intense activity levels (e.g. St. Patricks 2015 storm) the spatial gradients are important at very low L-shells (below the slot region), while during moderate or quiet conditions may span several L-shells up to L=4.5. Since there only a handful of events during Solar cycle 24 which are intense enough, one can understand that the amount of data-points at L<3 are significantly less and would introduce statistical errors in the binning process and, thus, the final statistics.

Concerning the error due to the spatial gradients at high L-shells we have already shown that it is negligible.

We have also revised the corresponding paragraph in the "Data and methods" section as follows:
"Note that, as the satellites move inbound and outbound with high velocities at low L-shells, the magnetic field measurements exhibit, not only orders of magnitude increase but, very large gradients as well. Therefore, the spatial variations of the field at low L-shells can no longer be removed due to the filtering process, thus, we remove them manually."

Other comments:
l.116 " drift-averaged coefficients". This is a misleading statement that needs to be removed. Drift averaged means drift phase averaged. Thus, it implies both spatial (MLT) and temporal averages. The data products in the database are only averaged over 1 hour in UT, and they are not MLT averaged.

Our response: As already mentioned, we have replaced the sentence in line 113 "These drift-averaged radial diffusion coefficients" with "These grouped radial diffusion coefficients".

l.157: "the uncertainty in DLL will be rather small": please provide quantification to support this claim, or reformulate.

Our response: We believe that we have already quantified this in figure 2 of the previous response to the reviewers. Nevertheless, we include figure 2 here as well.

[Figure]

The figure illustrates the distribution of the mean PSD versus the Beq/Blocal ratio for different values of L* from all three THEMIS spacecraft. As shown, both in the electric and magnetic field power, the variation of power versus Bratio is up to a factor of 2, at least for Bratio values larger than 0.8. We highlight that this 0.8 threshold in Bratio is used in our study (line 155-156 of the revised manuscript) so all results correspond to DLL values at points with Bratio>0.8.

REFERENCES
• Ali, A. F., D. M. Malaspina, S. R. Elkington, A. N. Jaynes, A. A. Chan, J.Wygant, and C. A. Kletzing: Electric and magnetic radial diffusion coefficients using the Van Allen probes data. J. Geophys. Res. Space Physics, 121, 9586–9607, doi:10.1002/2016JA023002, 2016.
• Drozdov, A. Y., Allison, H. J., Shprits, Y. Y., Elkington, S. R., and Aseev, N. A.: A comparison of radial diffusion coefficients in 1-D and 3-D long-term radiation belt simulations. Journal of Geophysical Research: Space Physics, 126, e2020JA028707, doi: 10.1029/2020JA028707, 2021.

- Falthammar, C.-G.: Effects of time-dependent electric fields on geomagnetically trapped radiation. J. Geophys. Res., 70(11), 2503‑2516, doi:10.1029/JZ070i011p02503, 1965.
- Fei, Y., A. A. Chan, S. R. Elkington, and M. J. Wiltberger: Radial diffusion and MHD particle simulations of relativistic electron transport by ULF waves in the September 1998 storm. J. Geophys. Res., 111, A12209, doi:10.1029/2005JA011211, 2006.
- Finlay, C.C., Kloss, C., Olsen, N., Hammer, M. Toeffner-Clausen, L., Grayver, A and Kuvshinov, A. (2020), The CHAOS-7 geomagnetic field model and observed changes in the South Atlantic Anomaly, Earth Planets and Space, 72, https://doi.org/10.1186/s40623-020-01252-9.
- Jaynes, A. N., Ali, A. F., Elkington, S. R., Malaspina, D. M., Baker, D. N., Li, X., Kanekal, S. G., Henderson, M. G., Kletzing, C. A., and Wygant, J. R.: Fast diffusion of ultrarelativistic electrons in the outer radiation belt: 17 March 2015 storm event. Geophysical Research Letters, 45(20), 10874–10882, doi:10.1029/2018GL079786, 2018.
- Lejosne, S., Kollmann, P. Radiation Belt Radial Diffusion at Earth and Beyond. Space Sci Rev 216, 19 (2020). https://doi.org/10.1007/s11214-020-0642-6
- Lejosne, S. (2020). Electromagneticradial diffusion in the Earth's radiation belts as determined by the solar wind immediate time history and a toy model for the electromagnetic fields. Journal of Geophysical Research: Space Physics, 125, e2020JA027893. https://doi.org/10.1029/2020JA027893
- Liu,W., W. Tu, X. Li, T. Sarris, Y. Khotyaintsev, 470 H. Fu, H. Zhang, and Q. Shi: On the calculation of electric diffusion coefficient of radiation belt electrons with in situ electric field measurements by THEMIS. Geophys. Res. Lett., 43, 1023–1030, doi:10.1002/2015GL067398, 2016.
- Olifer, L., Mann, I. R., Ozeke, L. G., Rae, I. J., and Morley, S. K.: On the relative strength of electric and magnetic ulf wave radial diffusion during the March 2015 geomagnetic storm. Journal of Geophysical Research: Space Physics, 124(4), 2569–2587. https://doi.org/10.1029/2018JA026348, 2019.
- Ozeke, L. G., I. R. Mann, K. R. Murphy, I. J. Rae, and D. K. Milling: Analytic expressions for ULF wave radiation belt radial diffusion coefficients. J. Geophys. Res. Space Physics, 119, 1587–1605, doi:10.1002/2013JA019204, 2014.
- Sandhu, J. K., Rae, I. J., Wygant, J. R., Breneman, A. W., Tian, S., Watt, C. E. J., et al. (2021). ULF wave driven radial diffusion during geomagnetic storms: A statistical analysis of Van Allen Probes observations. Journal of Geophysical Research: Space Physics, 126, e2020JA029024. https://doi.org/10.1029/2020JA029024

**Reviewer #2 Evaluations**:

General comments: The authors have implemented all the revisions requested, and significantly improved the structure and clarity of the discussions. The results are interesting and important, and I recommend this paper for prompt publication. If possible, I recommend the following grammatical / typographical revisions, but I think these can be implemented in the copyediting stage and should not hold up publication.

Recommended grammatical / typographical revisions:

Line 95: "quite similar with" Change to "quite similar to"

Line 103: "field measurements exhibit, not only…" Delete comma, change to: "field measurements exhibit not only…"

Line 107: "Especially for the magnetic component…" Delete "Especially", change to" "For the magnetic component"…

Line 348: "the relativistic electron population and but generally…" Delete "and", change to: "the relativistic electron population, but generally…"

Line 361: "dependence of these calculated DLL to several solar wind…" Replace "to" with "on", change to: "dependence of these calculated DLL on several solar wind…"

Our response: We thank the reviewer for noticing these errors, which we have now corrected.

---

## Author Response (AR3)

**Reviewer Evaluations**:

The major comments made during the previous rounds of reviews remain valid. Namely, the method chosen for data analysis is not supported by the physics of radiation belt radial diffusion. Thus, the data products presented are not convincing estimates of radial diffusion magnitude.

1. As previously stated: That the database presents significant variations with MLT is enough to demonstrate that the data products are not radial diffusion coefficients. It also shows that the objective of providing "event-specific" or "time-series" of radial diffusion coefficients using solely in-situ spacecraft measurements is unrealistic. Yet, this is what the manuscript proposes. The suggestions to remove the time series of DLLs and/or to rename the time series using a word other than "DLL" have been dismissed. I can't think of any other suggestion                    at                    this                    point. Discussing DLL as a function of MLT is inconsistent with the definition of a radial diffusion coefficient. In this context, I maintain that Figure 4 should be removed, together with any discussion    of    DLL    variations    with    MLT.    Yet,    this    has    not    been    done.

**Our response:**  We believe that there has been a misunderstanding here and we would like to further clarify this. The values shown in figure 4 (now figure 2 in the revised manuscript) do not correspond to the **final** hourly database values. The values used in this figure are the 1-min resolution proxy of the DLL at each point of the spacecraft orbit, for each spacecraft separately. Since this DLL proxy, at each L* value, has been calculated as the product of the weighted averaged power with a simple multiplication factor it is expected to reflect directly the azimuthal distribution of wave power for both the magnetic and the electric component. We are sorry if this was not fully clear previously and in order to explicitly state that in the manuscript we have included a separate section (section 3) where we use these DLL proxies (explicitly stated as proxies and not DLL) in order to discuss the possible uncertainties introduced by the limited MLT coverage of THEMIS and any other attempt to calculate DLL time-series using in-situ measurements.

However, as stated in the manuscript, our hourly DLL calculations are derived using the simultaneous measurements from all three THEMIS spacecraft.  Depending on the evolution of the azimuthal positions of the spacecraft, within the hourly time-bin we use, this results in an MLT coverage of up to ~6 hours, for each L* value. As such, there can be no MLT value associated with our hourly DLL, which are the final database products, as THEMIS spacecraft cover a wide range of azimuthal positions over one hour. **Therefore, even though our DLL calculations have uncertainties generated by the use of in-situ data, they do not violate – in any way – the physical definition of radial diffusion.** On the contrary, we have taken measures in order to minimize other significant uncertainties such as the use of the weighted averaged power, which minimizes the significant errors introduced by neglecting higher m (azimuthal wave mode) values.

We again emphasize that our DLL calculations include averaging from the three spacecraft which cover many different azimuthal positions over one hour, both individually and as a constellation. Therefore, the assumption that this partial azimuthal coverage can account for the entire drift of the electrons is exactly that, an assumption, and in no way a violation of the physical definition of the DLL. As we have thoroughly discussed in the previous responses these limitations are present in several recent works that use DLL time-series, e.g. Jaynes et al. (2018); Olifer et al. (2019); Sandhu et al. (2021). Another important example regarding such inherent limitations and necessary assumptions is the well-established Ozeke et al. (2014) semi-empirical model. It is notable that in the Ozeke model the electric

component of the DLL is inferred based on measurements from ground-based magnetometers and there the authors have used electric field estimations **only from dayside measurements**. One could argue that this is not only an obviously partial azimuthal coverage, but also that it introduces a very consistent and potentially important bias in the estimation of the DLL. However, despite such limitations the Ozeke model is well-established and widely used because it is one of the more well-performing semi-empirical models.

The aforementioned discussion, combined with our several arguments in the previous responses, oblige us to retain both the term radial diffusion coefficient and its symbol.

2. Going back to comments relative to the data processing: The manuscript still does not quantify the error accompanying the radial diffusion estimate. Yet, in addition to (*) the use of Fei's formula and (*) the lack of knowledge for the repartition of the field variations along the drift shell at any given time, (*) other assumptions are also adding error, in particular, when it comes to extracting "fast" field variations from "slow" and "spatial" field variations. Specifically:

- The reply considers that oscillations with frequencies higher than 0.8 mHz "are the ones responsible for the breaking of the third adiabatic invariant of relativistic electrons". Yet, this omits the fact that the frequency threshold should depend on the population energy. Indeed, the time variations of interest are the ones "faster than the trapped particles' drift period", and the drift period is energy dependent. In this context, I would also recommend detailing why the PSD of the waves considered is in the "2-25 mHz" in the manuscript (l.99).

**Our response:** We have included in lines 98-100 of the revised manuscript that the 2-25 mHz range correspond to near-equatorial mirroring electrons roughly in the 0.4 – 13 MeV energy range.

- In addition, it is claimed that spatial field variations are usually slow variations, and thus, are filtered out by the 20-min moving average. Yet, this is not proven. The illustration provided discusses ~50 min (?) of LEO B field data (from Swarm) to advocate for the lack of importance of spatial averaging. Yet, this does not prove that a 20-min temporal averaging is similar to spatial averaging for the fields measured along THEMIS orbits (which appears to be the assumption made in the manuscript). If the proof can be made for THEMIS, I would recommend adding the proof to the manuscript to justify this aspect of the approach.

**Our response:** The CHAOS model is developed for the Swarm data and, therefore, any attempt to apply it to other instruments would require inter-calibration of the measurements, something that is well beyond the scope of this work (even though we intend to enrich our database in the future with other missions as well). As Swarm spacecraft are in LEO, we showcase here that as an approach, qualitatively the application of a high pass-filter is adequate to remove the variations due to the satellite flying through an inhomogeneous magnetic field, even in LEO, where **the magnetic field gradients are extremely steep.** Therefore, we safely argue that if it is adequate for LEO, then it is more than adequate for data at much higher altitudes (such as those from THEMIS used here), where the magnetic gradients are significantly less steep.

At very low L values, steep gradients are removed "manually" (l.108). Yet, it is not explained how.

**Our response:** After the application of the filtering, we perform a visual inspection of the time series and the steep gradients that have not been removed by 20-min averaging are removed by hand.

The work also lacks a discussion for the conditions of validity for the electric and magnetic field measurements for THEMIS (e.g., for the E field: Califf et al., 2015, doi:10.1002/2014JA020360).

**Our response:** The uncertainties by the use of in-situ instruments have been included in line 150 of the revised manuscript.

**Response to Editor**:

The thinking behind this recommendation is that in this way you can retain your results (diffusion coefficient dependencies, database etc.), while in the same time provide some caution to the readers/users of your database on what kind of (systematic) errors may be associated with your coefficient estimation assumptions. The reviewer has some additional recommendations for discussing error/uncertainty estimates which may be of help. In addition, that may prevent from having a work, which reviewers (also ref. #2 of the original report) find interesting and original, rejected partly on the basis of terminology and difficult to avoid assumptions.

Dear Editor,

First of all we want to thank you for all the effort you have put in this manuscript. We further want to clarify a certain issue that may have been the source of a serious misunderstanding.

We have never treated the DLL as "local". We have explicitly stated in section 2.1 that the DLL is calculated as hourly average of the three THEMIS spacecraft, which provides us with an MLT coverage up to ~6 hours per hour and per L* (line 150 of the revised manuscript). Therefore, there is no MLT dependence of the DLL included in the database and which is shown in our results. On the other hand, in figure 4 only (now figure 2 in the revised manuscript), we do not show the same DLL, but rather a "local DLL" which reflects exactly the azimuthal distribution of power. We understand that this was not explicitly clarified in the previous responses and, to that end, we have included a new section (section 3 in the revised manuscript) where we use these local proxies in order to discuss the uncertainties introduced by the limited MLT coverage of in-situ data. We emphasize that this "local DLL" is explicitly referred as "DLL proxy" throughout the entire section.

Therefore, we feel appropriate and correct to retain both the symbol and the term DLL in our manuscript.

---

## Author Response (AR4)

**Reviewer Evaluations**:

Review of the article

I do not endorse the approach presented in this manuscript, and the scientific rigor of the data analysis could be improved. That said, I will not oppose the publication of this work provided that the assumptions and data analysis steps are clearly presented. On that point, I do not have major comments anymore.

Even when so, the manuscript content could still be improved, as discussed below. In particular, there are still some inaccurate and/or missing points:

l.108: "we remove them manually": please specify, in the main text, what is done here. As of now, a reader could not reproduce this approach. This could also be a good place to mention the difficulty to differentiate between spatial and temporal field variations.
**Our response:** We have added a brief description in section 2.1 (lines 106-108) and 2.3 (lines 149-152).

l. 123: "this assumption denounces the very concept of stochastic acceleration restricting the process to a resonant interaction". This claim needs to be substantiated. It is inconsistent with the first works on radial diffusion (e.g., Parker, 1960, https://doi.org/10.1029/JZ065i010p03117; Falthammar, 1965, https://doi.org/10.1029/JZ070i011p02503)
**Our response:** We have erased the sentence in order to avoid any misinterpretations. The text now (Line 123) is as follows: "This assumption can lead to underestimation of the radial diffusion coefficient, since higher m values are shown to be often significant (e.g. m=2 up to m=5 at recovery phase of storms".

l.137: "by a factor 2": this statement is true only in a special case, and not in general. See Lejosne (2019): "This result relies on the assumptions that (1) the magnetic field disturbances are described by the simple model introduced by Fälthammar (1965) (equations 2 and 3) and that (2) there is no electric potential disturbance."
**Our response:** Duly amended (lines 136-138).

l.186, Fig.2: The increase in DLL_E in the 1-2 MLT sector could also be due to electric field measurement errors at times when the spacecraft are in the Earth's shadow. Has this effect been considered?
**Our response:** We agree with the reviewer that this could also be a possible explanation. However, we would expect to see similar pronounced increases in pre-midnight MLTs as well, something that is not evident.

l.198 : "the 6 hours MLT coverage by the three THEMIS spacecraft" This statement needs to be proven or reformulated.
**Our response:** The sentence (lines 197-200) has been modified as follows: "This means that the partial azimuthal coverage provided by the three THEMIS spacecraft could lead to an up to one order of magnitude of uncertainty in the estimation of the DLLB for particular spatial configurations of the three THEMIS spacecrafts, e.g. when all three are located in the nightside or all three are located in the dayside."

l.290-292: These statements consist of putting the work presented here as the reference work. This is misleading. In particular, the conclusion statement l. 386-388 are very

misleading and it should be reformulated. It is not because the DLL coeffs provided by this analysis are greater than most DLLs than "all models underestimate DLL".

**Our response:** We don't understand how these sentences are misleading or setting our work as reference. Concerning the statements in lines 290-292, we have emphasized that: "the terms overestimation / underestimation of the DLL by the available semi-empirical models are always in comparison with our calculated values, which of course are themselves estimations of the true radial diffusion coefficient". This is a common procedure when comparing different datasets or, in our case, a dataset inferred from in situ measurements with models (lines 291-292). Nevertheless, we have modified the text in order to remove the terms overestimation / underestimation. At the same extend, we have modified the text at the conclusions section.

Section 6.2: it is not explained how, from a partial coverage in time and L* for the DLL database (limited to THEMIS orbits), a database covering all times and L* is obtained to be used in the simulation. How often are the DLL values updated? And why?

**Our response:** Obviously, the database contains gaps both in L* and in time due to the orbit of the THEMIS mission. For the St. Patricks event, the limited amount of gaps was filled using interpolation/extrapolation schemes. In detail, the gaps in the DLL spectra with respect to L* were interpolated using power law interpolation/extrapolation, and then we performed a time interpolation at each L* value (a description has been added in lines 355-356). Concerning the update of the DLL values we are not sure what the reviewer means. The calculations have been performed once and the output cdf files have been stored in the url we have provided. If the reviewer means update to include further temporal coverage, this is something we intend to do in the future.